EMBO
Molecular Medicine

# SARS-CoV-2 worldwide replication drives rapid rise and selection of mutations across the viral genome: a time-course study – potential challenge for vaccines and therapies

Stefanie Weber[1,†], Christina M Ramirez[2,†], Barbara Weiser[3], Harold Burger[3] & Walter Doerfler[1,4,*]

## Abstract

Scientists and the public were alarmed at the first large viral variant of SARS-CoV-2 reported in December 2020. We have followed the time course of emerging viral mutants and variants during the SARS-CoV-2 pandemic in ten countries on four continents. We examined > 383,500 complete SARS-CoV-2 nucleotide sequences in GISAID (Global Initiative of Sharing All Influenza Data) with sampling dates extending until April 05, 2021. These sequences originated from ten different countries: United Kingdom, South Africa, Brazil, United States, India, Russia, France, Spain, Germany, and China. Among the 77 to 100 novel mutations, some previously reported mutations waned and some of them increased in prevalence over time. VUI2012/01 (B.1.1.7) and 501Y.V2 (B.1.351), the so-called UK and South Africa variants, respectively, and two variants from Brazil, 484K.V2, now called P.1 and P.2, increased in prevalence. Despite lockdowns, worldwide active replication in genetically and socio-economically diverse populations facilitated selection of new mutations. The data on mutant and variant SARS-CoV-2 strains provided here comprise a global resource for easy access to the myriad mutations and variants detected to date globally. Rapidly evolving new variant and mutant strains might give rise to escape variants, capable of limiting the efficacy of vaccines, therapies, and diagnostic tests.

**Keywords** high incidence of C to T transitions; numerous new mutations; South African and Brazil variants; time course of SARS-CoV-2 mutant emergence; UK variant B.1.1.7

**Subject Categories** Chromatin, Transcription & Genomics; Microbiology, Virology & Host Pathogen Interaction

## Introduction

Between December 2019 and January 28, 2021, the severe acute respiratory syndrome coronavirus 2 (SARS-CoV-2) pandemic has expanded worldwide to 219 countries and territories; about 101.9 million people have been infected, and about 2.2 million (2.16%) have lost their lives according to Johns Hopkins (Dong *et al*, 2020). Note added in proof: As of May 04, 2021, 154.4 million COVID-19 cases and 3.23 million fatalities (2.09%) have been reported worldwide (https://www.worldometers.info/coronavirus/).

In our laboratory, we have set out to follow the rapid rise of new mutations in the SARS-CoV-2 genome as COVID-19 cases soared worldwide. We identified mutation hotspots in different populations. Initially, we analyzed SARS-CoV-2 sequences that had been deposited in databases between January and May/June of 2020. At least 10 prevalent sites of sequence mutations were observed and up to 80% of nucleotides at the mutated site had been exchanged (Weber *et al*, 2020). Several of these mutations led to non-synonymous amino acid changes in different open reading frames across the viral genome. These alterations in functional viral proteins were selected during active worldwide replication of SARS-CoV-2. We have now extended the time frame of mutant analyses to January 20 and for that of variants further to March 31, 2021 and found increased prevalence of mutations along the genome worldwide. We specifically examined mutations from the United States, India, Brazil, Russia, the UK, France, Spain, Germany, South Africa, and China that were deposited in the GISAID (Global Initiative of Sharing All Influenza Data) database (Elbe & Buckland-Merritt, 2017).

As of January 28, 2021, infection rates worldwide were extremely high, surpassing the levels seen at the peak in April 2020 (Dong *et al*, 2020). The uncontrolled spread has led to a proliferation of mutants and variants, which we define as viruses with a specific set of mutations. The so-called UK variant, also known as B.1.1.7 or alternatively VOC202012/01, was first identified in England in

1  Institute for Clinical and Molecular Virology, Friedrich-Alexander University (FAU), Erlangen, Germany
2  Department of Biostatistics, UCLA School of Public Health, Los Angeles, CA, USA
3  Department of Medicine, University of California, Davis, Sacramento, CA, USA
4  Institute of Genetics, University of Cologne, Cologne, Germany
   *Corresponding author. Tel: +49 171 205 1587; E-mail: walter.doerfler@t-online.de
   †These authors contributed equally to this work

September 2020 and reported on December 8 as a rapidly spreading variant of concern that had 14 mutations in total and three deletions (for details, see Table 1) (https://virological.org/t/preliminary-genomic-characterisation-of-an-emergent-sars-cov-2-lineage-in-the-uk-defined-by-a-novel-set-of-spike-mutations/563). Some of the mutations involve the gene for the Spike protein, which mediates binding, fusion, and entry of the virus into the host cell. One of these deletions, H69/V70 del (ΔH69/ΔV70), has been reported to emerge during convalescent plasma treatment (preprint: Kemp et al, 2021). Another Spike mutation, N501Y, is of concern, has been suggested to interact with ACE2, and could reduce the effectiveness of neutralizing antibodies (Yi et al, 2020). This variant has been associated with higher transmissibility (https://khub.net/documents/135939561/338928724/SARS-CoV-2+variant+under+investigation%2C+meeting+minutes.pdf/962e866b-161f-2fd5-1030-32b6ab467896; Volz et al, 2021) and at least one confirmed case of reinfection (Harrington et al, 2021) leading to lockdowns and travel bans in efforts to contain its spread. On December 23, 2020, the time of the lockdown, the variant was already found in Australia, Denmark, and Italy. As of April 5, 2021, this variant has been reported in 108 countries according to GISAID (https://www.gisaid.org/hcov19-variants) (Table 2).

On December 18, 2020, another variant of concern, unrelated to the UK variant but also having the N501Y mutation, was announced in South Africa and was dubbed 501Y.V2 or B.1.351 (Tegally et al, 2021). This variant is characterized by eight mutations in the Spike including K417N, E484K, and N501Y (https://virological.org/t/a-preliminary-selection-analysis-of-the-south-african-v501-v2-sars-cov-2-clade/573; Tegally et al, 2021) (Table 1). As of January 29, 2021, this variant has been reported in 68 countries and five continents.

Also rising independently are two Brazil variants that are now called P.1 and P.2. P.1 that have 17 unique amino acid changes, three deletions, four synonymous mutations, and one 4 nucleotide insertion (preprint: Faria et al, 2021) (Table 1). P.1 shares the N501Y and a deletion in ORF1ab with both the UK and the South Africa variant. It is interesting to note that the N501Y mutation was not widely spread in Brazil before this variant was described while the E484K is more prevalent, although Brazil is not sequencing large numbers of samples. The E484K and the N501Y mutations are of particular concern in that they have been suggested to reduce neutralization by antibodies and increase the affinity for ACE2. P.1 and B.1.351 share both mutations N501Y and E484K (Table 1). P.1 has been associated with a case of documented reinfection (https://

**Table 1. Mutations associated with variants B.1.1.7 (UK), B.1.135 (South Africa), P.1 (Brazil), P.2 (Brazil), B.1.525 (New York), B.1.526 (New York), B.1.427 (California), and B.1.429 (California).**

| Gene | B. 1. 1.7 Mutation | B.1.135 Mutation | P.1 Mutation | P.2 Mutation | B.1.525 Mutation | B.1.526 Mutation | B.1.427 Mutation | B.1.429 Mutation |
|---|---|---|---|---|---|---|---|---|
| ORF1ab | T1001I | | | | P314F | P314L | L452R | S13I |
| | A1708D | | | | T2007O | Q1011H | D614G | W152C |
| | I2230T | | | | | T265I | | L452R |
| | | | | | | L3201P | | D614G |
| | SGF 3675-3677 del | SGF 3675-3677 del | SGF 3675-3677 del | | | 3575-3677 del | | |
| nsp5 | | | | L205V | | | | |
| nsp6 | | | | | | | | |
| Spike | H69/V70 del | L18F | | | A67V | L5F* | | |
| | Y144 del | D80A | | | H68/V70del | T95I | | |
| | N501Y | D215G | | | Y144del | D253G | | |
| | A570D | R246I | | | | S477N* | | |
| | P681H | K417N | K417N | | | | | |
| | T716I | E484K | E484K | E484K | E484K | E484K* | | |
| | S982A | N501Y | N501Y | | D614G | | | |
| | D118H | A701Y | | V1176F | Q677H | *not in all sequences | | |
| | | | | | | F888L | | |
| Orf8 | Q27stop | | | | | | | |
| | R52I | | | | | | | |
| | Y73C | | | | | | | |
| Nucleocapsid | D3L | | | | A119S | A12G | | |
| | S235F | | | | R203K | T205I | | |
| | | | | | G204R | | | |
| | | | | | M234I | | | |

Table 2. B.1.1.7, B.1.351, P.1, B.1.427 + B.1.429, B.1.525: Variants of concern/interest of SARS-CoV-2 by country as of March 31, 2021. Currently, new variants are being detected and characterized in rapid succession. This Table could be outdated by the time of publication. For updating of data, consult GISAID (Shu & McCauley, 2017).

| Country | B.1.1.7 | B.1.351 | P.1 | B.1.429 & B.1.427 | B.1.525 |
|---|---|---|---|---|---|
| Albania | 28 | 0 | 0 | 0 | 0 |
| Angola | 6 | 7 | 0 | 0 | 1 |
| Argentina | 2 | 0 | 0 | 1 | 0 |
| Aruba | 120 | 2 | 1 | 31 | 0 |
| Australia | 242 | 38 | 4 | 17 | 8 |
| Austria | 414 | 167 | 0 | 2 | 3 |
| Bangladesh | 10 | 19 | 0 | 0 | 0 |
| Barbados | 3 | 0 | 0 | 0 | 0 |
| Belarus | 1 | 0 | 0 | 0 | 0 |
| Belgium | 5,302 | 655 | 223 | 1 | 24 |
| Bonaire | 91 | 0 | 0 | 0 | 0 |
| Bosnia and Herzegovina | 21 | 0 | 0 | 0 | 0 |
| Botswana | 0 | 54 | 0 | 0 | 0 |
| Brazil | 71 | 1 | 641 | 0 | 0 |
| British Virgin Islands | 0 | 0 | 0 | 1 | 0 |
| Brunei | 0 | 1 | 0 | 0 | 0 |
| Bulgaria | 659 | 0 | 0 | 0 | 0 |
| Cambodia | 7 | 0 | 0 | 2 | 0 |
| Cameroon | 0 | 1 | 0 | 0 | 1 |
| Canada | 2,395 | 38 | 150 | 13 | 13 |
| Cayman Islands | 2 | 0 | 0 | 0 | 0 |
| Chile | 30 | 0 | 42 | 10 | 0 |
| China | 14 | 1 | 0 | 0 | 0 |
| Colombia | 0 | 0 | 23 | 1 | 0 |
| Comoros | 0 | 6 | 0 | 0 | 0 |
| Costa Rica | 4 | 2 | 0 | 3 | 1 |
| Cote d'Ivoire | 7 | 0 | 0 | 0 | 4 |
| Croatia | 352 | 7 | 0 | 0 | 0 |
| Curacao | 107 | 0 | 0 | 0 | 0 |
| Cyprus | 10 | 0 | 0 | 0 | 0 |
| Czech Republic | 863 | 8 | 0 | 0 | 0 |
| Democratic Republic of the Congo | 2 | 1 | 0 | 0 | 0 |
| Denmark | 4,889 | 12 | 0 | 25 | 121 |
| Dominican Republic | 4 | 0 | 0 | 0 | 0 |
| Ecuador | 14 | 0 | 0 | 0 | 0 |
| England | 1 | 0 | 0 | 0 | 0 |
| Estonia | 273 | 3 | 0 | 0 | 0 |

Table 2 (continued)

| Country | B.1.1.7 | B.1.351 | P.1 | B.1.429 & B.1.427 | B.1.525 |
|---|---|---|---|---|---|
| Eswatini | 0 | 20 | 0 | 0 | 0 |
| Faroe Islands | 0 | 0 | 1 | 0 | 0 |
| Finland | 400 | 9 | 0 | 1 | 4 |
| France | 6,290 | 537 | 38 | 4 | 30 |
| French Guiana | 4 | 0 | 8 | 0 | 0 |
| Gambia | 3 | 0 | 0 | 0 | 0 |
| Georgia | 2 | 0 | 0 | 0 | 0 |
| Germany | 21,038 | 652 | 63 | 6 | 123 |
| Ghana | 116 | 4 | 0 | 0 | 6 |
| Gibraltar | 131 | 0 | 0 | 0 | 0 |
| Greece | 70 | 0 | 0 | 0 | 0 |
| Guadeloupe | 9 | 1 | 0 | 3 | 2 |
| Guam | 0 | 0 | 0 | 7 | 0 |
| Hungary | 29 | 0 | 0 | 0 | 0 |
| Iceland | 20 | 0 | 0 | 0 | 0 |
| India | 151 | 15 | 0 | 0 | 17 |
| Indonesia | 10 | 34 | 0 | 0 | 0 |
| Iran | 1 | 65 | 0 | 0 | 0 |
| Ireland | 4,583 | 39 | 11 | 0 | 16 |
| Israel | 1,769 | 0 | 0 | 7 | 0 |
| Italy | 6,909 | 0 | 394 | 1 | 73 |
| Jamaica | 4 | 0 | 0 | 0 | 0 |
| Japan | 456 | 22 | 25 | 17 | 11 |
| Jordan | 50 | 2 | 3 | 0 | 2 |
| Kenya | 20 | 37 | 0 | 0 | 0 |
| Kosovo | 3 | 0 | 0 | 0 | 0 |
| Kuwait | 1 | 0 | 0 | 0 | 0 |
| Latvia | 150 | 0 | 0 | 0 | 0 |
| Lebanon | 2 | 0 | 0 | 0 | 0 |
| Lesotho | 0 | 14 | 0 | 0 | 0 |
| Lithuania | 413 | 5 | 0 | 0 | 0 |
| Luxembourg | 669 | 180 | 3 | 0 | 1 |
| Malawi | 1 | 152 | 0 | 0 | 0 |
| Malaysia | 3 | 9 | 0 | 0 | 2 |
| Martinique | 6 | 0 | 0 | 0 | 0 |
| Mauritius | 1 | 2 | 0 | 0 | 0 |
| Mayotte | 1 | 378 | 0 | 0 | 1 |
| Mexico | 33 | 0 | 5 | 146 | 0 |
| Moldova | 3 | 0 | 0 | 0 | 0 |
| Monaco | 1 | 1 | 0 | 0 | 0 |
| Montenegro | 7 | 0 | 0 | 0 | 0 |
| Morocco | 1 | 0 | 0 | 0 | 0 |
| Mozambique | 0 | 58 | 0 | 0 | 0 |
| Namibia | 0 | 9 | 0 | 0 | 0 |

Table 2 (continued)

| Country | B.1.1.7 | B.1.351 | P.1 | B.1.429 & B.1.427 | B.1.525 |
|---|---|---|---|---|---|
| Netherlands | 6,854 | 341 | 59 | 5 | 36 |
| New Zealand | 98 | 23 | 4 | 4 | 0 |
| Nigeria | 128 | 0 | 0 | 0 | 0 |
| North Macedonia | 60 | 0 | 0 | 1 | 106 |
| Northern Mariana Islands | 0 | 0 | 0 | 1 | 0 |
| Norway | 1,630 | 190 | 1 | 2 | 22 |
| Oman | 1 | 0 | 0 | 0 | 0 |
| Pakistan | 7 | 0 | 0 | 0 | 0 |
| Panama | 0 | 1 | 0 | 0 | 0 |
| Paraguay | 0 | 0 | 5 | 0 | 0 |
| Peru | 3 | 0 | 23 | 0 | 0 |
| Philippines | 39 | 0 | 0 | 0 | 0 |
| Poland | 1,987 | 10 | 0 | 0 | 9 |
| Portugal | 1,701 | 48 | 20 | 0 | 3 |
| Reunion | 0 | 16 | 0 | 0 | 0 |
| Romania | 191 | 1 | 2 | 0 | 0 |
| Russia | 11 | 3 | 0 | 0 | 0 |
| Rwanda | 3 | 11 | 0 | 0 | 5 |
| Saint Lucia | 9 | 0 | 0 | 0 | 0 |
| Senegal | 3 | 0 | 0 | 0 | 0 |
| Serbia | 2 | 0 | 0 | 0 | 0 |
| Singapore | 88 | 71 | 0 | 4 | 3 |
| Sint Maarten | 27 | 0 | 1 | 13 | 30 |
| Slovakia | 609 | 7 | 0 | 0 | 0 |
| Slovenia | 839 | 25 | 1 | 0 | 0 |
| South Africa | 1 | 1,670 | 0 | 0 | 0 |
| South Korea | 103 | 5 | 1 | 47 | 1 |
| Spain | 4,352 | 31 | 20 | 2 | 18 |
| Sri Lanka | 19 | 1 | 0 | 0 | 1 |
| Sweden | 4,290 | 296 | 15 | 2 | 0 |
| Switzerland | 5,134 | 125 | 29 | 4 | 9 |
| Taiwan | 5 | 6 | 0 | 7 | 0 |
| Thailand | 12 | 0 | 0 | 0 | 1 |
| Togo | 2 | 1 | 0 | 0 | 0 |
| Trinidad and Tobago | 1 | 0 | 0 | 0 | 0 |
| Tunisia | 1 | 0 | 0 | 0 | 0 |
| Turkey | 522 | 112 | 5 | 2 | 12 |
| Ukraine | 22 | 0 | 0 | 0 | 0 |
| United Arab Emirates | 21 | 5 | 0 | 0 | 0 |
| United Kingdom | 187,267 | 434 | 31 | 16 | 275 |
| United States | 15,117 | 290 | 252 | 23,328 | 182 |

Table 2 (continued)

| Country | B.1.1.7 | B.1.351 | P.1 | B.1.429 & B.1.427 | B.1.525 |
|---|---|---|---|---|---|
| Vietnam | 11 | 0 | 0 | 0 | 0 |
| Zambia | 0 | 31 | 0 | 0 | 0 |
| Zimbabwe | 0 | 194 | 0 | 0 | 0 |

virological.org/t/sars-cov-2-reinfection-by-the-new-variant-of-concern-voc-p-1-in-amazonas-brazil/596), and 225 cases have been reported in the United States, and cases from 32 other countries have been deposited into GISAID. P.2, unrelated to P.1, is characterized by the E484K mutation and has been implicated in two cases of reinfection (Nonaka et al, 2021; https://virological.org/t/spike-e484k-mutation-in-the-first-sars-cov-2-reinfection-case-confirmed-in-brazil-2020/584). Analysis of samples in Southern California led to the identification of the "California variant" (Zhang et al, 2021) also known as B.1.429 or B.1.427 (https://www.cdph.ca.gov/Programs/CID/DCDC/Pages/COVID-19/COVID-Variants.aspx) depending on the pattern of mutations. Table 1 describes the pattern of mutations. The New York variant was described during the same time period (preprint: Annavajhala et al, 2021; preprint: West et al, 2021), although it is not deemed a variant of concern yet. The B.1.525 was also found in New York and is a variant of interest (https://www.cdc.gov/coronavirus/2019-ncov/cases-updates/variant-surveillance/variant-info.html).

These variants have caused concerns regarding efficacy of the vaccines. Recently, preprint: Wang et al (2021) described the efficacy of mRNA-1273 vaccine against many spike mutations tested both separately and in combination. They show that sera from both vaccinated non-human primates and vaccinated humans are effective against the UK variant and various other spike mutations. They also found neutralization, albeit at lower levels, against the full South Africa variant B.1.135. It has been shown that the Pfizer BNT162b2 vaccine is effective against the N501Y mutant alone (Xie et al, 2021) as well as the UK variant B.1.117 (Collier et al, 2021). There have also been preliminary data from two other vaccine manufacturers showing efficacy against the South African variant. To illustrate the rise of mutations and variants over time, we list the number of variants and mutations deposited in GISAID worldwide across time (Figure 1). Table 2 lists the number of variant sequences deposited in GISIAD by country.

The rapid appearance of the variants across the world illustrates the importance of sequencing viral pathogens and tracking mutations. There is emerging evidence that these variants may alter transmissibility and have the potential to reduce the efficacy of existing COVID-19 vaccines. Sequencing SARS-CoV-2 is both a scientific and clinical imperative (https://www.cogconsortium.uk/wp-content/uploads/2021/01/Report-2_COG-UK_SARS-CoV-2-Mutations.pdf). Because nucleic acid sequencing of SARS-CoV-2 samples is not part of routine clinical practice at this time, it is necessary to institute programs to monitor sequence variation as a matter of course in order to detect mutations in the viral genome.

A consequence of the lack of routine viral sequencing is that it may contribute to selection bias. Sequences deposited to GISAID may not be representative of viral prevalence as different countries contribute different numbers of sequences. It is also possible that selection bias may be inherent, as different countries deposit

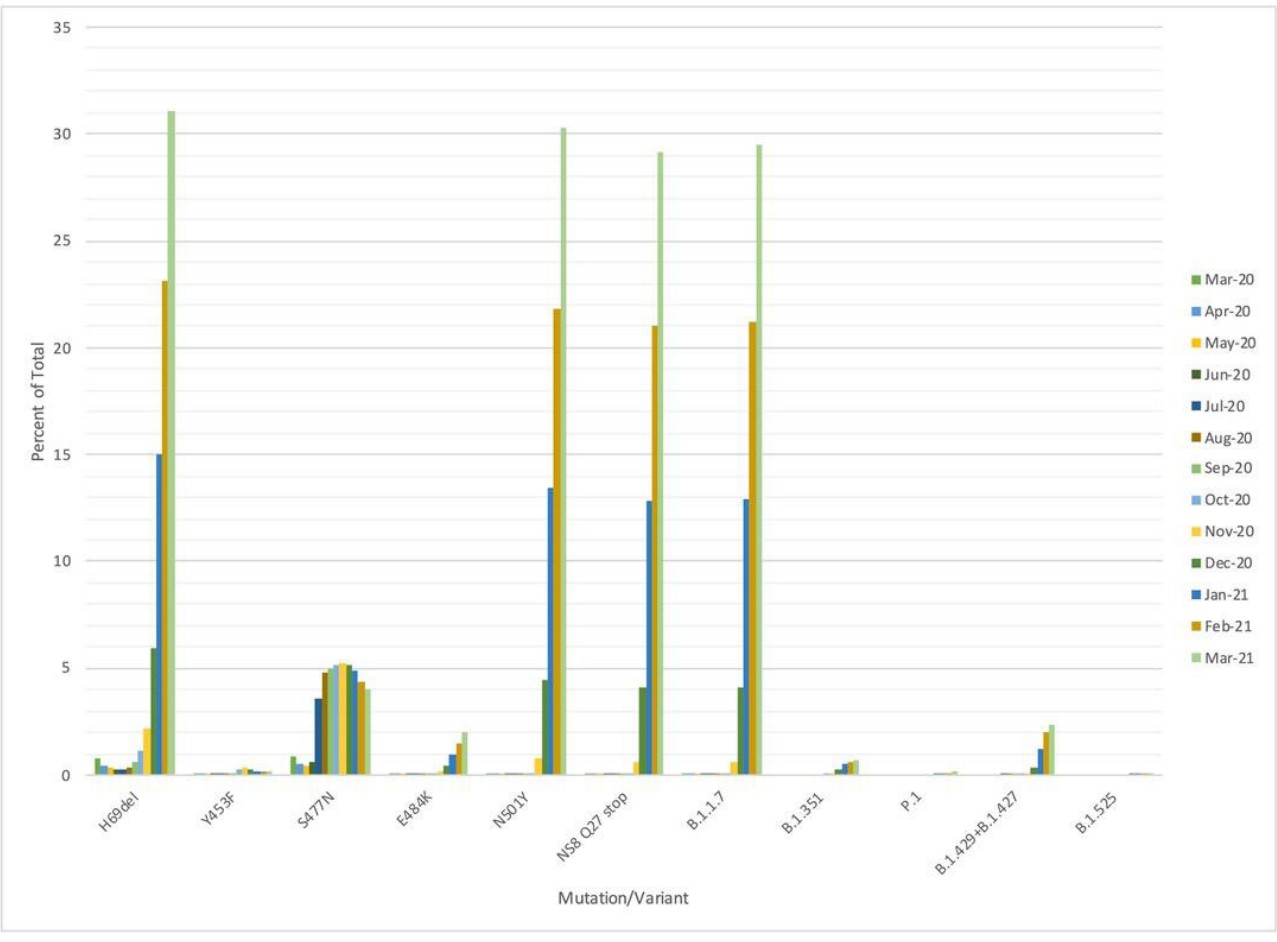

**Figure 1.** Relative proportions of mutations and variants of concern deposited to GISAID as of March 31. Time course study.

sequences at different rates and often not at random. It may be the case that more interesting samples or those deemed more likely to be a variant are preferentially sequenced. This is a likely case for samples that are selected for sequencing due to SGTF (spike gene target failure). It has been found that the Spike ΔH69/ΔV70 causes the so-called S dropout, rendering the nucleic acid test (NAT) negative for Spike (S) and positive for nucleocapsid (N). As this is one of the mutations in B.1.1.7, it has been used as a screening tool for this variant (preprint: Washington *et al*, 2021). While useful for screening, this deletion might create selection bias because patients who were positive for SARS-CoV-2 with an S dropout may have their samples preferentially sequenced as the prevalence for the new variant is being assessed.

Rapid increases in the number and types of new SARS-CoV-2 mutations in the world population within a time span of weeks to months are a remarkable biologic event. The uncontrolled rapid replication of SARS-CoV-2 in an immunologically naïve world population since early 2020 constituted a wake-up call of the need to sequence and track the evolution of novel pathogens as these mutations and variants have raised concerns regarding increased transmissibility, immune escape, and the efficacy of vaccines and the validity of diagnostic tests.

## Results

### Time course of emerging mutations in ten different countries

We examined mutations in 383,570 complete sequences with known sampling dates in GISAID up until January 20, 2021. Figure 1 shows the worldwide distribution of Spike mutations as well as other variants of interest over time from April 2020 to March 31, 2021, from complete sequences with a known collection date deposited in GISAID. Table 1 lists the signature mutations for the variants. Table 2 shows the total number of complete sequences each variant of interest (B.1.1.7 (the UK variant), 501Y.V2 (the South African variant) and 484K.V2 (B.1.1 lineage with S: E484K/D614G, V1176F N: A199S/R203K/G204R) deposited in GISAID by each country as of March 31, 2021.

Selection of novel mutations in humans was rapid and frequent in 2020. Among the novel mutations discovered in the current study, some were seen only in one country and others occurred in several different countries. We will present the identified mutations arising in the SARS-CoV-2 RNA country by country for the designated time periods (Tables 3–12). The data covering time course analyses of the appearance of mutations and their nature in most of the ten different countries are presented in Tables 3A–12A. The corresponding B

**Table 3. United Kingdom.**

| Position | Location | Mutation | Total Count | Percentage |
|---|---|---|---|---|
| | | | 01/19/2020–01/20/2021 | |
| 66nt | 5′UTR | C → T | 2,787 | 3.9 |
| 204nt | | G → T | 20,770 | 29.07 |
| 241nt | | C → T | 69,160 | 96.81 |
| 445nt | ORF1ab polyprotein → leader protein | T → C | 34,505 | 48.3 |
| 1,163nt | nsp2 | A → T | 2,544 | 3.56 |
| 1,210nt | | G → T | 1,440 | 2.02 |
| 1,513nt | | C → T | 1,528 | 2.14 |
| 1,947nt | | T → C | 1,576 | 2.21 |
| 1,987nt | | A → G | 3,018 | 4.22 |
| 3,037nt | nsp3 | C → T | 69,231 | 96.91 |
| 3,256nt | | T → C | 2,523 | 3.53 |
| 4,002nt | | C → T | 1,519 | 2.13 |
| 4,543nt | | C → T | 1,516 | 2.12 |
| 6,286nt | | C → T | 34,650 | 48.5 |
| 6,807nt | | C → T | 2,220 | 3.11 |
| 7,528nt | | C → T | 1,524 | 2.13 |
| 7,926nt | | C → T | 2,818 | 3.94 |
| 8,683nt | nsp4 | C → T | 2,189 | 3.06 |
| 9,745nt | | C → T | 3,640 | 5.1 |
| 9,802nt | | G → T | 1,449 | 2.03 |
| 10,097nt | 3C-like proteinase | G → A | 2,954 | 4.13 |
| 10,870nt | | G → T | 3,186 | 4.46 |
| 11,083nt | nsp6 | G → T | 5,734 | 8.03 |
| 11,396nt | | C → T | 2,286 | 3.2 |
| 11,533nt | | A → G | 1,960 | 2.74 |
| 11,781nt | | A → G | 2,368 | 3.31 |
| 12,067nt | nsp7 | G → T | 1,709 | 2.39 |
| 13,536nt | RNA-dependent RNA polymerase | C → T | 1,502 | 2.1 |
| 14,202nt | | G → T | 2,522 | 3.53 |
| 14,408nt | | C → T | 69,237 | 96.92 |
| 14,805nt | | C → T | 1,860 | 2.6 |
| 15,406nt | | G → T | 2,077 | 2.91 |
| 18,877nt | 3′-to-5′ exonuclease | C → T | 3,827 | 5.36 |
| 19,542nt | | G → T | 2,582 | 3.61 |
| 19,718nt | endoRNAse | C → T | 2,645 | 3.7 |
| 20,268nt | | A → G | 1,999 | 2.8 |
| 21,255nt | 2′-O-ribose methyltransferase | G → C | 34,494 | 48.28 |
| 21,575nt | Spike glycoprotein | C → T | 1,502 | 2.1 |
| 21,614nt | | C → T | 17,561 | 24.58 |
| 21,637nt | | C → T | 2,697 | 3.78 |

**Table 3** (continued)

| Position | Location | Mutation | Total Count | Percentage |
|---|---|---|---|---|
| | | | 01/19/2020–01/20/2021 | |
| 22,227nt | | C → T | 34,855 | 48.79 |
| 22,346nt | | G → T | 2,244 | 3.14 |
| 22,377nt | | C → T | 1,518 | 2.12 |
| 22,388nt | | C → T | 2,540 | 3.56 |
| 22,444nt | | C → T | 2,085 | 2.92 |
| 22,992nt | | G → A | 1,636 | 2.29 |
| 23,403nt | | A → G | 69,262 | 96.95 |
| 23,731nt | | C → T | 2,940 | 4.12 |
| 24,334nt | | C → T | 10,442 | 14.62 |
| 25,563nt | ORF3a | G → T | 5,774 | 8.08 |
| 25,614nt | | C → T | 2,737 | 3.83 |
| 26,060nt | | C → T | 2,632 | 3.68 |
| 26,144nt | | G → T | 1,748 | 2.45 |
| 26,424nt | Envelope protein | T → C | 1,957 | 2.74 |
| 26,735nt | Membrane glycoprotein | C → T | 3,760 | 5.26 |
| 26,801nt | | C → G | 34,459 | 48.24 |
| 27,769nt | ORF7b | C → T | 2,706 | 3.79 |
| 27,944nt | ORF8 | C → T | 25,177 | 35.24 |
| 28,169nt | | A → G | 2,693 | 3.77 |
| 28,854nt | Nucleocapsid phosphoprotein | C → T | 3,683 | 5.16 |
| 28,881nt | | G → A | 23,975 | 33.56 |
| 28,882nt | | G → A | 23,947 | 33.52 |
| 28,883nt | | G → C | 23,946 | 33.52 |
| 28,932nt | | C → T | 34,536 | 48.34 |
| 29,227nt | | G → T | 2,566 | 3.59 |
| 29,366nt | | C → T | 1,743 | 2.44 |
| 29,466nt | | C → T | 2,578 | 3.61 |
| 29,555nt | At upstream downstream region of ORF10 ORF9 | C → T | 1,466 | 2.05 |
| 29,645nt | ORF10 | G → T | 34,684 | 48.55 |
| 29,771nt | 3′UTR | A → G | 2,475 | 3.46 |

Details of the mutant analyses of 7,144 SARS-CoV-2 isolates for deviations from the Wuhan reference sequence. These sequences were deposited in the GISAID initiative between 01/19/2020 and 01/20/2021. For design of Tables, see legend to Table 5.

Tables summarize the total number of mutations in individual sequence position at a cutoff of 2% preponderance for the time period 01/19/2020 to 01/20/2021, i.e., of the entire first COVID-19 year. Of course, it can be argued that a cutoff for the registration of mutants at 2% incidence is arbitrary. However, we cannot predict with certainty which mutations at low incidence of occurrence at present will become more predominant in the future during rapid worldwide viral replication in the current pandemic. A feasible

**Table 4. South Africa.**

| (A) Position | Location | Mutation | 09/01–12/07/2020 Count | Incidence |
|---|---|---|---|---|
| 174nt | 5′UTR | GT → TT | 12/95 | DE,US |
| | | noneffective | | |
| 241nt | | CG → TG | 95/95 | Prevalent |
| | | noneffective | | |
| 1,059nt | nsp2 | CC → TC | 10/95 | Prevalent |
| | | A**CC** (Threonine) → A**TC** (Isoleucine) | | |
| 2,164nt | | GA → CA | 11/95 | IN |
| | | GA**GA**AG (Glutamic Acid Lysine) → GA**CA**AG (Aspartic Acid Lysine) | | |
| 3,037nt | nsp3 | CT → TT | 95/95 | Prevalent |
| | | noneffective | | |
| 5,230nt | | GT → TT | 12/95 | DE |
| | | AA**GT**GG (Lysine Tryptophan) → AA**TT**GG (Asparagine Tryptophan) | | |
| 6,762nt | | CT → TT | 13/95 | Unique |
| | | A**CT** (Threonine) → A**T**T (Isoleucine) | | |
| 10,323nt | 3C-like proteinase | AG → GG | 11/95 | Unique |
| | | A**AG** (Lysine) → A**GG** (Arginine) | | |
| 11,230nt | nsp6 | GC → TC | 11/95 | Unique |
| | | AT**GC**CT (Methionine Proline) → AT**TC**CT (Isoleucine Proline) | | |
| 12,503nt | nsp8 | TA → CA | 26/95 | Unique |
| | | **TA**T (Tyrosine) → **CA**T (Histidine) | | |
| 14,408nt | RNA-dependent RNA polymerase | CT → TT | 95/95 | Prevalent |
| | | C**CT** (Proline) → C**TT** (Leucine) | | |
| 20,268nt | endoRNAse | AG → GG | 21/95 | FR,ES,RU |
| | | noneffective | | |
| 21,801nt | Spike glycoprotein | AT → CT | 10/95 | Unique |
| | | G**AT** (Aspartic Acid) → G**CT** (Alanine) | | |
| 22,675nt | | CG → TG | 10/95 | Unique |
| | | noneffective | | |
| 22,813nt | | GA → TA | 10/95 | DE |
| | | noneffective | | |
| 23,012nt | | GA → AA | 12/95 | IN |
| | | **GA**A (Glutamic Acid) → **AA**A (Lysine) | | |
| 23,403nt | | AT → GT | 95/95 | Prevalent |
| | | G**AT** (Aspartic Acid) → G**GT** (Glycine) | | |
| 23,664nt | | CA → TA | 14/95 | ES,IN |
| | | G**CA** (Alanine) → G**TA** (Valine) | | |
| 25,563nt | ORF3a protein | GA → TA | 10/95 | Prevalent |
| | | CA**GA**GC (Glutamine Serine) → CA**TA**GC (Histidine Serine) | | |
| 25,770nt | | GC → TC | 20/95 | RU |
| | | AG**GC**TT (Arginine Leucine) → AG**TC**TT (Serine Leucine) | | |
| 25,904nt | | CA → TA | 10/95 | BR,DE |
| | | T**CA** (Serine) → T**TA** (Leucine) | | |

**Table 4** (continued)

| (A) Position | Location | Mutation | 09/01–12/07/2020 Count | Incidence |
|---|---|---|---|---|
| 26,456nt | Envelope protein | CT → TT | 10/95 | Unique |
| | | C**CT** (Proline) → C**TT** (Leucine) | | |
| 28,253nt | ORF8 protein | CA → TA | 14/95 | BR,DE,ES,FR,US |
| | | noneffective | | |
| 28,854nt | Nucleocapsid phosphoprotein | CA → TA | 23/95 | CN,DE,ES,FR,IN,RU |
| | | T**CA** (Serine) → T**TA** (Leucine) | | |
| 28,881nt | | GGG → AAC | 61/95 | Prevalent |
| | | A**GGG**GA (Arginine Glycine) → A**AAC**GA (Lysine Arginine) | | |
| 28,887nt | | CT → TT | 11/95 | BR,CN,FR,IN,RU |
| | | A**CT** (Threonine) → A**TT** (Isoleucine) | | |
| 29,721 | 3′UTR | CC → TC | 26/95 | Unique |
| | | noneffective | | |

| (B) Position | Location | Mutation | 01/19/2020–01/20/2021 Total Count | Percentage |
|---|---|---|---|---|
| 174nt | 5′UTR | G → T | 181 | 10.17 |
| 241nt | | C → T | 1,772 | 99.61 |
| 355nt | ORF1ab polyprotein → leader protein | C → T | 59 | 3.32 |
| 1,059nt | nsp2 | C → T | 149 | 8.38 |
| 2,094nt | | C → T | 38 | 2.14 |
| 2,164nt | | G → C | 84 | 4.72 |
| 2,692nt | | A → T | 41 | 2.3 |
| 3,037nt | nsp3 | C → T | 1,746 | 98.15 |
| 4,002nt | | C → T | 165 | 9.27 |
| 4,093nt | | C → T | 48 | 2.7 |
| 5,230nt | | G → T | 147 | 8.26 |
| 6,027nt | | C → T | 46 | 2.59 |
| 6,762nt | | C → T | 178 | 10.01 |
| 7,064nt | | A → G | 124 | 6.97 |
| 8,660nt | nsp4 | C → T | 69 | 3.88 |
| 8,964nt | | C → T | 69 | 3.88 |
| 9,498nt | | T → C | 36 | 2.02 |
| 10,097nt | 3C-like proteinase | G → A | 163 | 9.16 |
| 10,323nt | | A → G | 169 | 9.5 |
| 11,083nt | nsp6 | G → T | 60 | 3.37 |
| 11,230nt | | G → T | 75 | 4.22 |
| 11,447nt | | G → A | 129 | 7.25 |
| 12,503nt | nsp8 | T → C | 389 | 21.87 |
| 13,536nt | RNA-dependent RNA polymerase | C → T | 170 | 9.56 |
| 14,408nt | | C → T | 1,773 | 99.66 |
| 14,925nt | | C → T | 71 | 3.99 |
| 16,376nt | Helicase | C → T | 54 | 3.04 |
| 16,490nt | | C → T | 39 | 2.19 |
| 16,853nt | | G → T | 47 | 2.64 |
| 16,946nt | | C → T | 43 | 2.42 |

**Table 4** (continued)

| (B) | | | 01/19/2020–01/20/2021 | |
|---|---|---|---|---|
| Position | Location | Mutation | Total Count | Percentage |
| 18,747nt | 3'-to-5' exonuclease | C → T | 115 | 6.46 |
| 20,234nt | endoRNAse | C → T | 42 | 2.36 |
| 20,268nt | | A → G | 209 | 11.75 |
| 21,801nt | Spike glycoprotein | A → C | 142 | 7.98 |
| 22,206nt | | A → G | 71 | 3.99 |
| 22,287nt | | T → A | 86 | 4.83 |
| 22,299nt | | G → T | 69 | 3.88 |
| 22,675nt | | C → T | 290 | 16.3 |
| 22,813nt | | G → T | 139 | 7.81 |
| 23,012nt | | G → A | 146 | 8.21 |
| 23,063nt | | A → T | 140 | 7.87 |
| 23,403nt | | A → G | 1,772 | 99.61 |
| 23,625nt | | C → T | 53 | 2.98 |
| 23,664nt | | C → T | 154 | 8.66 |
| 23,731nt | | C → T | 161 | 9.05 |
| 25,455nt | ORF3a | G → T | 65 | 3.65 |
| 25,521nt | | C → T | 66 | 3.71 |
| 25,563nt | | G → T | 148 | 8.32 |
| 25,770nt | | G → T | 285 | 16.02 |
| 25,904nt | | C → T | 143 | 8.04 |
| 26,456nt | Envelope protein | C → T | 140 | 7.87 |
| 26,586nt | Membrane glycoprotein | C → T | 62 | 3.49 |
| 27,384nt | ORF6 | T → C | 120 | 6.75 |
| 27,504nt | ORF7a | T → C | 50 | 2.81 |
| 28,077nt | ORF8 | G → T | 74 | 4.16 |
| 28,253nt | | C → T | 178 | 10.01 |
| 28,854nt | Nucleocapsid phosphoprotein | C → T | 173 | 9.72 |
| 28,881nt | | G → A | 1,238 | 69.59 |
| 28,882nt | | G → A | 1,238 | 69.59 |
| 28,883nt | | G → C | 1,238 | 69.59 |
| 28,887nt | | C → T | 152 | 8.54 |
| 29,425nt | | G → T | 117 | 6.58 |
| 29,721nt | 3'UTR | C → T | 388 | 21.81 |

The Table presents characteristics of SARS-CoV-2 mutants from South African isolates. For Table design, see legend to Table 5.

strategy will be to install mutant watch programs and remain on the alert for the rise of new mutations. This strategy can be implemented only by highly efficient SARS-CoV-2 RNA sequencing strategies that will have to be instituted as widely as possible and without delay.

**Mutation analyses in ten different countries**

The following paragraphs document the mutational repertoire of SARS-CoV-2 in different regions of the world. The results are somewhat biased in that countries differed considerably in the number of sequences that had become available for inspection in the GISAID database (www.gisaid.org) (Shu & McCauley, 2017). We have

emphasized the time course of appearance of novel mutations in SARS-CoV-2 isolates that had a history of vigorous replication in some of the most severely affected populations on the globe, such as UK, South Africa, the United States, India, Brazil, Russia, France, Spain, Germany, and China. The most recent update [January 30, 2021] of COVID-19 cases and fatalities in the ten countries, whose isolates were analyzed for mutations, is presented in Table 13.

**United Kingdom**

For mutations arising in the UK, we have not followed the time course of emerging mutations during earlier periods of the

**Table 5.   United States.**

| (A) Position | Location | Mutation | 02/29–04/26/2020* Count | 06/12–07/07/2020* Count | 07/09–07/22/2020 Count | 08/01–12/01/2020 Count | Incidence |
|---|---|---|---|---|---|---|---|
| 241nt | 5′UTR | CG → TG | 76/111 | 74/96 | 99/99 | 116/117 | Prevalent |
| | | noneffective | | | | | |
| 1,059nt | nsp2 | CC → TC | 42/112 | 45/97 | 30/99 | 56/117 | Prevalent |
| | | A**CC** (Threonine) → A**TC** (Isoleucine) | | | | | |
| 1,917nt | | CT → TT | 0/112 | 11/97 | 0/99 | 0/117 | CN |
| | | A**CT** (Threonine) → A**TT** (Isoleucine) | | | | | |
| 2,416nt | | CA → TA | 9/112 | 4/97 | 1/99 | 3/117 | CN,ES,FR,RU, ZA |
| | | noneffective | | | | | |
| 3,037nt | nsp3 | CT → TT | 75/112 | 72/97 | 99/99 | 117/117 | prevalent |
| | | noneffective | | | | | |
| 3,871nt | | GA → TA | 0/112 | 0/97 | 29/99 | 4/117 | FR,ZA |
| | | AA**GA**TC (Lysine Isoleucine) → AA**TA**TC (Asparagine Isoleucine) | | | | | |
| 3,931nt | | TG → CG | 0/112 | 0/97 | 29/99 | 4/117 | Unique |
| | | noneffective | | | | | |
| 4,226nt | | CC → TC | 0/112 | 0/97 | 28/99 | 0/117 | Unique |
| | | **CC**A (Proline) → **TC**A (Serine) | | | | | |
| 5,672nt | | CC → TC | 0/112 | 0/97 | 28/99 | 0/117 | Unique |
| | | **CC**T (Proline) → **TC**T (Serine) | | | | | |
| 7,837nt | | AG → CG | 0/112 | 0/97 | 28/99 | 0/117 | CN |
| | | TT**AG**AC (Leucine Aspartic Acid) → TT**CG**AC (Phenylalanine Aspartic Acid) | | | | | |
| 8,083nt | | GG → AG | 0/112 | 0/97 | 0/99 | 18/117 | Unique |
| | | AT**GG**AA (Methionine Glutamic Acid) → AT**AG**AA (Isoleucine Glutamic Acid) | | | | | |
| 8,782nt | nsp4 | CC → TC | 15/112 | 15/97 | 0/99 | 0/117 | CN,DE,ES,IN |
| | | noneffective | | | | | |
| 10,139nt | 3C-like proteinase | CT → TT | 0/112 | 0/97 | 0/99 | 29/117 | Unique |
| | | **CT**T (Leucine) → **TT**T (Phenylalanine) | | | | | |
| 12,025nt | nsp7 | CA → TA | 0/112 | 0/97 | 11/99 | 2/117 | Unique |
| | | noneffective | | | | | |
| 14,408nt | RNA-dependent RNA polymerase | CT → TT | 78/112 | 71/97 | 99/99 | 117/117 | Prevalent |
| | | C**CT** (Proline) → C**TT** (Leucine) | | | | | |
| 17,747nt | Helicase | CT → TT | 8/112 | 12/97 | 0/99 | 0/117 | FR |
| | | C**CT** (Proline) → C**TT** (Leucine) | | | | | |
| 17,858nt | | AT → GT | 8/112 | 12/97 | 0/99 | 0/117 | ZA |
| | | T**AT** (Tyrosine) → T**GT** (Cysteine) | | | | | |
| 18,060nt | 3′- to −5′ exonuclease | CT → TT | 9/112 | 11/97 | 0/99 | 0/117 | ZA |
| | | noneffective | | | | | |
| 18,424nt | | AA → GA | 0/112 | 0/97 | 0/99 | 26/117 | Unique |
| | | **AA**T (Asparagine) → **GA**T (Aspartic Acid) | | | | | |
| 18,486nt | | CA → TA | 0/112 | 0/97 | 13/99 | 2/117 | Unique |
| | | noneffective | | | | | |

**Table 5** (continued)

| (A) Position | Location | Mutation | 02/29– 04/26/ 2020* Count | 06/12– 07/07/ 2020* Count | 07/09– 07/22/ 2020 Count | 08/01– 12/01/ 2020 Count | Incidence |
|---|---|---|---|---|---|---|---|
| 18,877nt | | CT → TT | 13/112 | 1/97 | 6/99 | 3/117 | BR,DE,ES,FR,IN |
| | | noneffective | | | | | |
| 19,677nt | endoRNAse | GG → TG | 0/112 | 0/97 | 26/99 | 0/117 | Unique |
| | | CA**GG**GT (Glutamine Glycine) → CA**TG**GT (Histidine Glycine) | | | | | |
| 19,839nt | | TA → CA | 0/112 | 0/97 | 11/99 | 7/117 | CN,DE,ES,FR, RU |
| | | noneffective | | | | | |
| 20,268nt | | AG → GG | 2/112 | 5/97 | 15/99 | 29/117 | FR,ES,RU,ZA |
| | | noneffective | | | | | |
| 21,304nt | 2'-O-ribose methyltransferase | CG → TG | 0/112 | 0/97 | 0/99 | 25/117 | ES |
| | | **CG**C (Arginine) → **TG**C (Cysteine) | | | | | |
| 22,162nt | Spike glycoprotein | TT → CT | 0/112 | 0/97 | 13/99 | 2/117 | Unique |
| | | noneffective | | | | | |
| 23,403nt | | AT → GT | 77/112 | 72/97 | 99/99 | 117/117 | Prevalent |
| | | G**AT** (Aspartic Acid) → G**GT** (Glycine) | | | | | |
| 23,707nt | | CA → TA | 0/112 | 0/97 | 11/99 | 3/117 | Unique |
| | | noneffective | | | | | |
| 25,907nt | ORF3a protein | GT → TT | 0/112 | 0/97 | 0/99 | 26/117 | Unique |
| | | G**GT** (Glycine) → G**TT** (Valine) | | | | | |
| 25,563nt | | GA → TA | 65/112 | 54/97 | 37/99 | 66/117 | Prevalent |
| | | CA**GA**GC (Glutamine Serine) → CA**TA**GC (Histidine Serine) | | | | | |
| 27,964nt | ORF8 protein | CA → TA | 13/112 | 6/97 | 4/99 | 31/117 | Unique |
| | | T**CA** (Serine) → T**TA** (Leucine) | | | | | |
| 28,144nt | | TA → CA | 15/112 | 15/97 | 0/99 | 0/117 | CN,DE,ES,IN |
| | | T**TA** (Leucine) → T**CA** (Serine) | | | | | |
| 28,472nt | Nucleocapsid phosphoprotein | CC → TC | 0/112 | 0/97 | 0/99 | 22/117 | Unique |
| | | **CC**T (Proline) → **TC**T (Serine) | | | | | |
| 28,821nt | | CT → AT | 0/112 | 0/97 | 9/99 | 5/117 | Unique |
| | | T**CT** (Serine) → T**AT** (Tyrosine) | | | | | |
| 28,854nt | | CA → TA | 3/112 | 0/97 | 13/99 | 28/117 | CN,DE,ES,FR, IN,RU |
| | | T**CA** (Serine) → T**TA** (Leucine) | | | | | |
| 28,869nt | | CA → TA | 0/112 | 0/97 | 0/99 | 25/117 | DE |
| | | C**CA** (Proline) → C**TA** (Leucine) | | | | | |
| 28,881nt | | GGG → AAC | 3/112 | 1/97 | 17/99 | 17/117 | Prevalent |
| | | A**GGG**GA (Arginine Glycine) → A**AAC**GA (Lysine Arginine) | | | | | |
| 28,887nt | | CT → TT | 0/112 | 1/97 | 1/99 | 10/117 | BR,CN,FR,IN, RU |
| | | A**CT** (Threonine) → A**TT** (Isoleucine) | | | | | |
| 28,977nt | | CT → TT | 0/112 | 0/97 | 29/99 | 4/117 | CN |
| | | T**CT** (Serine) → T**TT** (Phenylalanine) | | | | | |

**Table 5** (continued)

| (B) Position | Location | Mutation | 01/19/2020–01/20/2021 | |
|---|---|---|---|---|
| | | | Total Count | Percentage |
| 36nt | 5′UTR | C → T | 1,188 | 2.24 |
| 241nt | | C → T | 48,826 | 92.24 |
| 833nt | nsp2 | T → C | 1,171 | 2.21 |
| 1,059nt | | C → T | 28,844 | 54.49 |
| 3,037nt | nsp3 | C → T | 49,077 | 92.71 |
| 8,083nt | | G → A | 2,779 | 5.25 |
| 8,782nt | nsp4 | C → T | 2,798 | 5.29 |
| 10,319nt | 3C-like proteinase | C → T | 8,465 | 15.99 |
| 10,323nt | | A → G | 1,176 | 2.22 |
| 10,741nt | | C → T | 1,120 | 2.12 |
| 11,083nt | nsp6 | G → T | 1,612 | 3.05 |
| 11,916nt | nsp7 | C → T | 1,670 | 3.15 |
| 14,408nt | RNA-dependent RNA polymerase | C → T | 49,140 | 92.83 |
| 14,805nt | | C → T | 3,176 | 6 |
| 16,260nt | Helicase | C → T | 1,797 | 3.39 |
| 17,747nt | | C → T | 2,049 | 3.87 |
| 17,858nt | | A → G | 2,084 | 3.94 |
| 18,060nt | 3′-to-5′ exonuclease | C → T | 2,135 | 4.03 |
| 18,424nt | | A → G | 6,708 | 12.67 |
| 18,877nt | | C → T | 1,517 | 2.87 |
| 19,839nt | endoRNAse | T → C | 1,955 | 3.69 |
| 20,268nt | | A → G | 6,742 | 12.74 |
| 21,304nt | 2′-O-ribose methyltransferase | C → T | 6,603 | 12.47 |
| 23,403nt | Spike glycoprotein | A → G | 49,154 | 92.86 |
| 23,604nt | | C → A | 1,238 | 2.34 |
| 24,076nt | | T → C | 2,148 | 4.06 |
| 25,563nt | ORF3a | G → T | 31,241 | 59.02 |
| 25,907nt | | G → T | 6,369 | 12.03 |
| 27,964nt | ORF8 | C → T | 12,002 | 22.67 |
| 28,144nt | | T → C | 2,790 | 5.27 |
| 28,472nt | Nucleocapsid phosphoprotein | C → T | 6,473 | 12.23 |
| 28,821nt | | C → A | 1,821 | 3.44 |
| 28,842nt | | G → T | 1,152 | 2.18 |
| 28,854nt | | C → T | 6,694 | 12.65 |
| 28,869nt | | C → T | 6,640 | 12.54 |
| 28,881nt | | G → A | 6,887 | 13.01 |
| 28,882nt | | G → A | 6,848 | 12.94 |
| 28,883nt | | G → C | 6,847 | 12.93 |
| 28,887nt | | C → T | 1,090 | 2.06 |
| 29,402nt | | G → T | 1,630 | 3.08 |

**Table 5** (continued)

| (B) | | | 01/19/2020–01/20/2021 | |
| Position | Location | Mutation | Total Count | Percentage |
|---|---|---|---|---|
| 29,784nt | 3′UTR | C → T | 1,062 | 2.01 |
| 29,870nt | | C → A | 1,990 | 3.76 |

The general design of this Table is similar to Tables 3, 4 and 7–12, with minor modifications. Part A: From the overall analyses of the entire SARS-CoV-2 RNA sequence from 112 (US-I), 97 (US-II), 99 (US-III), and 117 (US-IV) randomly chosen isolates, the mutated nucleotides (nt)—as compared to the original Wuhan sequence—were tabulated. The actual time periods of mutant selections for the US-I to US-IV samples were indicated. Please note that in some of the Tables, as is the case in Table 5A, mutations were analyzed at different time intervals. From earlier to later, these time intervals were designated in the text as US-I, US-II, etc. The same nomenclature was followed in other Tables as well, in case more than one time interval was studied. Mutations previously designated as "signal hotspots" (Weber *et al*, 2020, i.e. 241–1,059–1,440–2,891–3,037–8,782–14,408–23,403–25,563–28,144–28,881) were now designated "prevalent." The * in the US-I and US-II columns designates previous publication in (Weber *et al*, 2020). The actual nucleotide changes were indicated in the third column, the most frequent being C → T (here 61.5%), as reported previously (Simmonds, 2020; Weber *et al*, 2020). Locations of mutations on the viral genome and amino acid exchanges as consequences of individual mutations were tabulated in columns 2 and 3, respectively. In columns 4 to 7, the actual frequencies of mutations at the four time intervals (US-1 to US-IV) are listed. The following designations for individual countries were chosen: BR for Brazil, CN for China, DE for Germany, FR for France, IN for India, RU for Russia, ES for Spain, ZA for South Africa, UK for United Kingdom, and US for United States.
The GGG → AAC is a non-point mutation in nucleotide position 28,881 that generated a highly basic amino acid sequence in the SARS-CoV-2 nucleocapsid phosphoprotein. We have speculated that this mutation might have originated from a recombination event between different viral RNA molecules (Weber *et al*, 2020).
Part B: A total of 5,710 SARS-CoV-2 RNA sequences from the GISAID source were analyzed. Deviations from the Wuhan reference sequence of >2% incidence were found at 42 sites in the sequence. Further details were described in the text.

pandemic. In a total of > 71,000 viral isolates of SARS-CoV-2 genomes from around the world, that were deposited between 01/19/2020 and 01/20/2021, four of the prevalent mutations found worldwide, at positions 241, 3,037, 14,408, and 23,403, had reached almost 100% representation (Table 3). In a total of 70 sequence positions > 2% deviations in comparison to the Wuhan reference were noted, > 50% were C to U (T) transitions (see also Tables 3–12B). Twelve novel mutations reached prevalence values between 15% and 49%, seven of them around 49%. Several of these mutations were also found in other countries (Tables 4–12). High prevalence of new mutations correlated with active replication in countries of high COVID-19 incidence.

On December 8, 2020, Rambaut *et al* described a novel variant of SARS-CoV-2 that was circulating in England starting in October and increased in prevalence suggesting a possible increase in transmissibility (https://virological.org/t/preliminary-genomic-characterisation-of-an-emergent-sars-cov-2-lineage-in-the-uk-defined-by-a-novel-set-of-spike-mutations/563; https://khub.net/documents/135939561/338928724/SARS-CoV-2 + variant + under + investigation%2C + meeting + minutes.pdf/962e866b-161f-2fd5-1030-32b6ab467896; https://www.cogconsortium.uk/wp-content/uploads/2021/01/Report-2_COG-UK_SARS-CoV-2-Mutations.pdf; Volz *et al*, 2021). An analysis of its genome revealed 14 non-synonymous mutations and 3 deletions that comprised a few nucleotides. In the spike glycoprotein, six of these mutations and two deletions were located, one of them N501Y due to an A23063T replacement. This particular variant is now considered a variant of concern VOC202012/01 (https://www.cogconsortium.uk/wp-content/uploads/2021/01/Report-2_COG-UK_SARS-CoV-2-Mutations.pdf). Current reports have described increased infectivity of this variant, whereas its pathogenicity is currently being assessed (Volz *et al*, 2021).

Recent reports suggest that the BioNTech/Pfizer BNT162b2 vaccine is effective against the UK variant as well as the N501Y mutant alone (Collier *et al*, 2021; Xie *et al*, 2021). Wu *et al* show preliminary effectiveness for the Moderna vaccine (mRNA-1273) (preprint: Wu *et al*, 2021) against the variant. Press reports from Novavax (https://ir.novavax.com/news-releases/news-release-details/novavax-covid-19-vaccine-demonstrates-893-efficacy-uk-phase-3) are also suggestive of the effectiveness of NVX-CoV2373 against the UK variant. Table 2 lists mutations found in the GISAID database up until March 31, 2021, and reports 187,267, 434, 31, 16, and 275 cases of variants B.1.1.7, B.1.351, P1, B.1.429 + B.1.427, and B.1.525, respectively.

### South Africa

We analyzed 95 SARS-CoV-2 sequences from viral isolates in South Africa that were deposited in the GISAID databank (Table 4A); 28 mutations overall were found in those sequences. Four of the seven prevalent mutations, known from isolates all over the world, had reached 100% representation in the SARS-CoV-2 sequences, except those at positions 1,059 (~ 10%), 25,563 (~ 10%), and 28,881 (~ 63%). There were seven new mutations unique to the South African isolates, four of which caused non-synonymous amino acid exchanges. Twelve of the novel mutations were shared with other countries, eight of these mutations led to amino acid exchanges, many of them to non-synonymous replacements. Twenty-five percent of the mutations affected the spike glycoprotein, a finding that should alert us to the capacity of the virus to respond to potential vaccines directed against the viral spikes. There was one each mutation that involved the viral endoRNAse and the RNA-dependent RNA polymerase.

For the entire year 2020 (January 19, 2020, to January 20, 2021), the four prevalent mutations at positions 241, 3,037, 14,408, and 23,403 were again (Table 4B) represented close to 100%, the mutation at 28,881/2/3 in the nucleocapsid phosphoprotein gene at about 70% (Table 4B). There were 8 new mutations at > 10% prevalence. In a total of 63 positions in the viral genome, deviations from the Wuhan reference sequence were noted above the 2% cutoff.

Recently, the N501Y variant was detected in South Africa which also had two additional point mutations, K417 and E484K. Data about its possible increased infectivity and transmissibility were preliminary (preprint: Cheng *et al*, 2021). Also in December 2020,

**Table 6. India.**

| (A) | | | 01/27–05/27/2020* | 06/03–07.04.2020 | |
|---|---|---|---|---|---|
| **Position** | **Location** | **Mutation** | **Count** | **Count** | **Incidence** |
| 241nt | 5´UTR | CG → TG | 82/99 | 95/98 | Prevalent |
| | | noneffective | | | |
| 2,292nt | nsp2 | AG → CG | 0/99 | 22/98 | Unique |
| | | C**AG** (Glutamine) → C**CG** (Proline) | | | |
| 2,836nt | nsp3 | CT → TT | 23/99 | 44/98 | Unique |
| | | noneffective | | | |
| 3,037nt | | CT → TT | 81/99 | 96/98 | Prevalent |
| | | noneffective | | | |
| 3,634nt | | CA → TA | 8/99 | 17/98 | ZA |
| | | noneffective | | | |
| 4,084nt | | CA → TA | 12/99 | 1/98 | ZA |
| | | noneffective | | | |
| 4,300nt | | GC → TC | 0/99 | 16/98 | Unique |
| | | noneffective | | | |
| 6,312nt | | CA → AA | 10/99 | 0/98 | US |
| | | A**CA** (Threonine) → A**AA** (Lysine) | | | |
| 11,083nt | nsp6 | GT → TT | 13/99 | 0/98 | BR,CN, DE, |
| | | TT**GT**AT (Leucine Tyrosine) → T**TT** (Phenylalanine) | | | ES,FR,US, |
| | | | | | ZA |
| 14,408nt | RNA-dependent RNA polymerase | CT → TT | 80/99 | 91/98 | Prevalent |
| | | C**CT** (Proline) → C**TT** (Leucine) | | | |
| 15,324nt | | CA → TA | 7/99 | 18/98 | B,C,G,F |
| | | noneffective | | | |
| 16,512nt | Helicase | AT → GT | 0/99 | 11/98 | Unique |
| | | noneffective | | | |
| 18,568nt | 3´- to - 5´exonuclease | CT → TT | 0/99 | 22/98 | Unique |
| | | **CT**C (Leucine) → **TT**C (Phenylalanine) | | | |
| 18,877nt | | CT → TT | 45/99 | 51/98 | BR,DE,ES, |
| | | noneffective | | | FR,US |
| 19,154nt | | CA → TA | 0/99 | 12/98 | Unique |
| | | A**CA** (Threonine) → A**TA** (Isoleucine) | | | |
| 21,724nt | Spike glycoprotein | GT → TT | 6/99 | 23/98 | RU |
| | | TT**GT**TC (Leucine Phenylalanine) → TT**TT**TC (Phenylalanine Phenylalanine) | | | |
| 22,444nt | | CC → TC | 26/99 | 48/98 | US |
| | | noneffective | | | |
| 23,403nt | | AT → GT | 80/99 | 96/98 | Prevalent |
| | | G**AT** (Aspartic Acid) → G**GT** (Glycine) | | | |
| 23,929nt | | CA → TA | 10/99 | 0/98 | FR,RU,US |
| | | noneffective | | | |
| 25,563nt | ORF3a protein | GA → TA | 43/99 | 51/98 | Prevalent |
| | | CA**GA**GC (Glutamine Serine) → CA**TA**GC (Histidine Serine) | | | |

**Table 6** (continued)

| (A) | | | 01/27–05/27/2020* | 06/03–07.04.2020 | |
|---|---|---|---|---|---|
| **Position** | **Location** | **Mutation** | **Count** | **Count** | **Incidence** |
| 26,735nt | Membrane glycoprotein | CA → TA | 39/99 | 49/98 | DE,ES,FR, |
| | | noneffective | | | US |
| 28,311nt | Nucleocapsid phosphoprotein | CC → TC | 10/99 | 0/98 | Unique |
| | | C**CC** (Proline) → C**TC** (Leucine) | | | |
| 28,854nt | | CA → TA | 29/99 | 41/98 | CN,DE,ES, |
| | | T**CA** (Serine) → T**TA** (Leucine) | | | FR,RU,US, |
| | | | | | ZA |

| (B) | | | 01/19/2020–01/20/2021 | |
|---|---|---|---|---|
| **Position** | **Location** | **Mutation** | **Total Count** | **Percentage** |
| 241nt | 5′UTR | C → T | 2,816 | 85.93 |
| 313nt | ORF1ab polyprotein → leader protein | C → T | 944 | 28.81 |
| 1,947nt | nsp2 | T → C | 100 | 3.05 |
| 2,292nt | | A → C | 73 | 2.23 |
| 2,836nt | nsp3 | C → T | 281 | 8.57 |
| 3,037nt | | C → T | 2,824 | 86.18 |
| 3,634nt | | C → T | 283 | 8.64 |
| 4,300nt | | G → T | 70 | 2.14 |
| 4,354nt | | G → A | 227 | 6.93 |
| 4,372nt | | A → G | 72 | 2.2 |
| 5,700nt | | C → A | 949 | 28.96 |
| 6,312nt | | C → A | 302 | 9.22 |
| 6,573nt | | C → T | 228 | 6.96 |
| 8,782nt | nsp4 | C → T | 74 | 2.26 |
| 8,917nt | | C → T | 122 | 3.72 |
| 9,693nt | | C → T | 156 | 4.76 |
| 11,083nt | nsp6 | G → T | 369 | 11.26 |
| 13,730nt | RNA-dependent RNA polymerase | C → T | 332 | 10.13 |
| 14,408nt | | C → T | 2,768 | 84.47 |
| 15,324nt | | C → T | 285 | 8.7 |
| 16,626nt | Helicase | C → T | 143 | 4.36 |
| 18,568nt | 3'-to-5' exonuclease | C → T | 71 | 2.17 |
| 18,877nt | | C → T | 654 | 19.96 |
| 19,524nt | | C → T | 69 | 2.11 |
| 21,550nt | 2'-O-ribose methyltransferase | A → C | 115 | 3.51 |
| 21,551nt | | A → T | 112 | 3.42 |
| 21,724nt | Spike glycoprotein | G → T | 109 | 3.33 |
| 22,444nt | | C → T | 507 | 15.47 |
| 22,468nt | | G → T | 76 | 2.32 |
| 23,403nt | | A → G | 2,832 | 86.42 |
| 23,929nt | | C → T | 298 | 9.09 |
| 25,528nt | ORF3a | C → T | 222 | 6.77 |
| 25,563nt | | G → T | 652 | 19.9 |
| 26,735nt | Membrane glycoprotein | C → T | 654 | 19.96 |

**Table 6** (continued)

| (B) | | | 01/19/2020–01/20/2021 | |
|---|---|---|---|---|
| **Position** | **Location** | **Mutation** | **Total Count** | **Percentage** |
| 27,384nt | ORF6 | T → C | 77 | 2.35 |
| 28,144nt | ORF8 | T → C | 73 | 2.23 |
| 28,311nt | Nucleocapsid phosphoprotein | C → T | 299 | 9.12 |
| 28,854nt | | C → T | 541 | 16.51 |
| 28,878nt | | G → A | 70 | 2.14 |
| 28,881nt | | G → A | 1,434 | 43.76 |
| 28,882nt | | G → A | 1,430 | 43.64 |
| 28,883nt | | G → C | 1,430 | 43.64 |
| 29,474nt | | G → T | 72 | 2.2 |
| 29,750nt | 3′UTR | C → T | 74 | 2.26 |
| 29,868nt | at downstream region of ORF10 | G → A | 351 | 10.71 |
| 29,870nt | | C → A | 154 | 4.7 |

The Table presents characteristics of SARS-CoV-2 mutants from isolates collected in the Indian population. For Table design, see legend to Table 5.

another variant called 501Y.V2, B.1.351 also known South African variant is characterized by eight lineage defining mutation with three in the receptor-binding domains: K417N, E484K, and N501Y. This variant also appeared to spread quickly in South Africa giving rise to travel bans from South Africa. It has been suggested that this variant is able to escape neutralization by donor plasma (Wibmer *et al,* 2021). Increased transmissibility has also been suggested (preprint: Cheng *et al,* 2021). Furthermore, there is early evidence that the efficacy of multiple existing vaccines against the B.1.351 variant may be diminished (https://www.janssen.com/johnson-johnson-announces-single-shot-janssen-covid-19-vaccine-candidate-met-primary-endpoints; https://ir.novavax.com/news-releases/news-release-details/novavax-covid-19-vaccine-demonstrates-893-efficacy-uk-phase-3; preprint: Wang *et al,* 2021). It will be important to continue to perform sequence analyses of viral strains and to correlate the evolution of mutants and variants with viral transmission and vaccine efficacy. As of March 2021, 1,670 cases of variant B.1.351 were reported in South Africa (Table 2).

## United States

Table 5A lists mutations from a random subset of sequences selected in the United States at 4 different time points. Some of the long-term prevalent mutations presented in the table under US-I and US-II were already included in a previous analysis as indicated by an asterisk (Weber *et al,* 2020). They were listed here again to facilitate comparisons to the wider spectrum of new mutations that arose in the United States (US-III, US-IV) and in different countries in the course of a few weeks. In addition to the worldwide occurring prevalent mutations, at nucleotide (nt) numbers 241, 1,059, 3,037, 8,782, 14,408, 23,403, 25,563, 28,144, and 28,881, there were a total of 13 unique, i.e., not previously described mutations in our analyses of which nine were found exclusively in the US-III sample cohort at frequencies between 4 and 29.3 % (Table 5A, unique). Except for three of these mutations, many attained their highest frequency of occurrence at the time point US-III. Two of the novel unique mutations in sequence positions 17,858 and

18,060 had disappeared in the US-III samples. Seventeen of the novel mutations were shared by other regions in the world, seven appeared in most or all ten countries investigated. We listed 13 mutations that had disappeared in the July samples of US-III, possibly they had proved not to be penetrating enough or were not sampled due to selection bias. As apparent in the table, five of the 15 new mutations among the US-II sequences deposited between June 12 and July 07 occurred at low frequencies (< 10%) exclusively in this collection of sequences, others, also at low frequencies, were also present in isolates from other countries as indicated. There were a number of novel shared mutations which were also represented in other countries—BR Brazil, CN China, FR France, DE Germany, IN India, RU Russia, ES Spain, and ZA South Africa. The more recently selected SARS-CoV-2 mutations under US-III stemmed from the time period between July 09 and July 22, 2020. The comparison of June and July US-III sequences and their mutations to their counterparts from a month earlier (US-II) revealed the complex vitality of new mutants arising in a SARS-CoV-2 population that had been replicating during a most critical phase of the US pandemic during the summer of 2020. During the four months' period 08/01 to 12/01 (US-IV), another 117 SARS-CoV-2 sequences were added to Table 5A. Several of the predominant mutations reached 100% representation. Eight novel mutations, some unique, others shared, were listed at nucleotide positions 8,083, 10,139, 18,424, 21,304, 25,907, 28,472, 28,869, and 28,887; most of them reached > 20% representation. At many nucleotide positions in the viral genome, the frequencies of the long-term predominant mutations increased over the entire time period between the last days of February to the end of July. This study has thus allowed us to witness the spread of mutations in the US population and at the same time the constant emergence of novel mutations and their increase in frequency with time.

## Impact on coding capacities

There is the idea that many mutations exist at low level, but are detected when they are selected and proliferate. Of the 39 SARS-

**Table 7. Brazil.**

| (A) | | | | 02/25–08/15/2020 | |
|---|---|---|---|---|---|
| Position | Location | Mutation | | Count | Incidence |
| 241nt | 5′UTR | CG → TG | | 95/101 | Prevalent |
| | | noneffective | | | |
| 3,037nt | nsp3 | CT → TT | | 97/102 | Prevalent |
| | | noneffective | | | |
| 12,053nt | nsp7 | CT → TT | | 16/102 | Unique |
| | | **CT**T (Leucine) → **TT**T (Phenylalanine) | | | |
| 14,408nt | RNA-dependent RNA polymerase | CT → TT | | 96/102 | Prevalent |
| | | C**CT** (Proline) → C**TT** (Leucine) | | | |
| 23,403nt | Spike glycoprotein | AT → GT | | 97/102 | Prevalent |
| | | G**AT** (Aspartic Acid) → G**GT** (Glycine) | | | |
| 25,088nt | | GT → TT | | 25/102 | Unique |
| | | **GT**T (Valine) → **TT**T (Phenylalanine) | | | |
| 27,299nt | ORF6 protein | TA → CA | | 41/102 | FR |
| | | A**TA** (Isoleucine) → A**CA** (Threonine) | | | |
| 28,881nt | Nucleocapsid phosphoprotein | GGG → AAC | | 73/102 | Prevalent |
| | | A**GGG**GA (Arginine Glycine) → A**AAC**GA (Lysine Arginine) | | | |
| 29,148nt | | TC → CC | | 41/100 | FR,RU |
| | | A**TC** (Isoleucine) → A**CC** (Threonine) | | | |

| (B) | | | 01/19/2020–01/20/2021 | |
|---|---|---|---|---|
| Position | Location | Mutation | Total Count | Percentage |
| 25nt | 5′UTR | T → A | 43 | 3.89 |
| 25nt | | T → G | 23 | 2.08 |
| 100nt | | C → T | 97 | 8.77 |
| 241nt | | C → T | 1,087 | 98.28 |
| 3,037nt | nsp3 | C → T | 1,093 | 98.82 |
| 3,766nt | | T → C | 49 | 4.43 |
| 6,319nt | | A → G | 32 | 2.89 |
| 10,667nt | 3C-like proteinase | T → G | 98 | 8.86 |
| 11,083nt | nsp6 | G → T | 29 | 2.62 |
| 11,824nt | | C → T | 98 | 8.86 |
| 12,053nt | nsp7 | C → T | 318 | 28.75 |
| 12,964nt | nsp9 | A → G | 89 | 8.05 |
| 14,408nt | RNA-dependent RNA polymerase | C → T | 1,091 | 98.64 |
| 23,012nt | Spike glycoprotein | G → A | 98 | 8.86 |
| 23,403nt | | A → G | 1,093 | 98.82 |
| 25,088nt | | G → T | 463 | 41.86 |
| 26,149nt | ORF3a | T → C | 31 | 2.8 |
| 27,299nt | ORF6 | T → C | 459 | 41.5 |
| 28,253nt | ORF8 | C → T | 110 | 9.95 |
| 28,628nt | Nucleocapsid phosphoprotein | G → T | 99 | 8.95 |
| 28,881nt | | G → A | 1,031 | 93.22 |
| 28,882nt | | G → A | 1,031 | 93.22 |
| 28,883nt | | G → C | 1,031 | 93.22 |

**Table 7** (continued)

| (B) | | | 01/19/2020–01/20/2021 | |
|---|---|---|---|---|
| Position | Location | Mutation | Total Count | Percentage |
| 28,975nt | | G → T | 101 | 9.13 |
| 29,148nt | | T → C | 466 | 42.13 |
| 29,754nt | 3′UTR | C → T | 95 | 8.59 |
| 29,861nt | | G → T | 33 | 2.98 |

The general design of these Tables follows the outline described in detail in the legend to Table 5 (United States). The number of sequences investigated for SARS-CoV-2 mutations is detailed in Tables for individual countries.

CoV-2 RNA sites mutated, 13 mutations, i.e., 42%, remained without effect on the encoded protein. In contrast, 18, i.e., 58%, exhibited changes in the genomes coding capacity [noted in bolded font in Table 5A] which affected most of the virus-encoded proteins. Most amino acid exchanges were non-synonymous and were likely responsible for functionally important alterations as judged from the type of amino acid replacements, e.g., pro to ser (nucleotide position 4,226) in nsp3; leu to phe (7,837), also in nsp3; tyr to cys (17,858) in the viral helicase; asp to gly (23,403) in the spike glycoprotein; arg-gly to lys-arg (28,881) in the nucleocapsid phosphoprotein and others. Among the additional eight mutations in the US-IV period, four led to non-synonymous amino acid exchanges in functionally important proteins as the 2'-O ribose-methyltransferase, the 5'-3' exonuclease, and the nucleocapsid phosphoprotein.

The asp to gly exchange due to the mutation in position 23,403 that affected the viral spike glycoprotein was described earlier (Korber *et al*, 2020). The mutant grows to higher titers in cell cultures and reaches higher viral loads in the upper respiratory tract but does not lead to increased disease severity (Korber *et al*, 2020). The mutation has been reported to increase susceptibility to neutralization. At this point, the functional consequences of most of the identified mutations for viral replication and/or pathogenicity need to be assessed. The SARS-Co-V-2 variant discovered in the UK in December 2020 will be discussed in part *(iii)* of the Discussion section.

### Analysis of mutation frequencies during short periods of time as compared to those observed over the entire year 2020

In addition, a total of 52,934 SARS-CoV-2 sequences from the United States in GISAID was analyzed for the presence of mutations as compared to the original Wuhan sequence (Table 5B) over the entire year 2020. A total of 42 sequence positions showed > 2% deviations from the reference sequence; 21 (50%) were C to U (T) transitions. Data from Table 5A indicate a C to U frequency of 61.5%. Similarly, high C to U preferences in sequence exchanges were observed in isolates from some of the other nine countries that were analyzed. In the Discussion section of this article, a presumptive editing function (APOBEC) is discussed to account for the prevalence of C to U transitions in all these viral genomes. SARS-CoV-2 represents itself as a highly adaptable virus that optimally utilizes its and the host cell's capacities to generated mutations and has them efficiently selected under a wide range of conditions in human populations.

As of March 31, 2021, the numbers of cases of variants B.1.1.7, B.1.351, P1, B.1.429 + B.1.427, and B.1.525 were reported to reach 15,117, 290, 252, 23,328, and 182, respectively (Table 2). Worldwide, the occurrence of SARS-CoV-2 mutations and variants is changing daily as expected at the height of this pandemic.

### India

During the periods of sequence analyses between January 27, 2020, to May 27, 2020 (IN-I), and June 03, 2020, to July 04, 2020 (IN-II), the prevalent hotspot mutations at sequence positions 241, 3,037, 14,408, 23,403, and 25,563 had reached values of representation approaching 100%, except at position 25,563 which amounted to 52% of sequences (Table 6A). New mutations emerged during these time periods. A set of nine novel mutations, unique to the Indian population, were observed, i.e., 39.1% out of a total of 23 mutations in all sub-samples from India.

These unique mutations were located in genome positions which were completely different from the newly arising SARS-CoV-2 mutations in the United States or in any other population investigated in our study (Table 6A). A total of seven of these novel mutations originated or increased in frequency in the late IN-II time period, whereas two of the mutations could no longer be detected during that same period. An additional nine newly arising mutations were shared with those in countries as indicated, some of which reached a frequency of up to 50%. Among all mutations from the Indian samples, C → U (T) transitions held the majority of 15/23, i.e., 65.2% (Table 13). We note that 18 out of 23 (78.3%) mutations in the SARS-CoV-2 isolates from our sub-samples from India were novel. About 7/9 of the India-unique mutations appeared *de novo* or increased in frequency within a time period of a few weeks of very active replication of the virus in the Indian population. New mutations are not only perpetually arising during the present stage of a nearly uncontrolled COVID-19 pandemic, but are also capable of becoming selected in the Indian population.

Table 6B lists 46 individual mutations for >3270 complete sequences with known sampling dates deposited to GISAID by January 20, 2021. The prevalent mutations at positions 241, 3,037, 14,408, and 23,403 (Tables 3–12) were represented at about 86%, at position 28,881 at 44%. In total, 46 positions showed mutations at frequency levels > 2%, 10 of them > 10%. The frequency of C to U transitions among all mutations in the samples from India was 50% (calculated from data in Table 6B).

As of March 31, 2021, the frequencies of variants of concern, B.1.1.7, B.1.351, P1, B.1.429 + B.1.427, and B.1.525 were reportedly 151, 15, 0, 0, and 17, respectively.

**Table 8. Russia.**

| (A) | | | 03/24–06/07/2020 | |
|---|---|---|---|---|
| **Position** | **Location** | **Mutation** | **Count** | **Incidence** |
| 241nt | 5′UTR | CG → TG | 215/226 | prevalent |
| | | noneffective | | |
| 3,037nt | nsp3 | CT → TT | 224/226 | prevalent |
| | | noneffective | | |
| 3,140nt | | CC → TC | 13/226 | unique |
| | | **CC**T (Proline) → AA**TC**TT (Asparagine Leucine) | | |
| 14,408nt | RNA-dependent RNA polymerase | CT → TT | 225/226 | prevalent |
| | | C**CT** (Proline) → C**TT** (Leucine) | | |
| 20,268nt | endoRNAse | AG → GG | 32/226 | ES,FR,US, |
| | | noneffective | | ZA |
| 23,403nt | Spike glycoprotein | AT → GT | 226/226 | prevalent |
| | | G**AT** (Aspartic Acid) → G**GT** (Glycine) | | |
| 25,563nt | ORF3a protein | GA → TA | 10/226 | prevalent |
| | | CA**GA**GC (Glutamine Serine) → CA**TA**GC (Histidine Serine) | | |
| 26,750nt | Membrane glycoprotein | CA → TA | 45/226 | unique |
| | | noneffective | | |
| 27,415nt | ORF6 protein | GC → TC | 10/226 | unique |
| | | **GC**A (Alanine) → **TC**A (Serine) | | |
| 28,881nt | Nucleocapsid phosphoprotein | GGG → AAC | 172/226 | prevalent |
| | | A**GGG**GA (Arginine Glycine) → A**AAC**GA (Lysine Arginine) | | |

| (B) | | | 01/19/2020–01/20/2021 | |
|---|---|---|---|---|
| **Position** | **Location** | **Mutation** | **Total Count** | **Percentage** |
| 30nt | 3′UTR | A → G | 57 | 4.75 |
| 241nt | | C → T | 1,167 | 97.33 |
| 1,059nt | nsp2 | C → T | 31 | 2.59 |
| 3,037nt | nsp3 | C → T | 1,188 | 99.08 |
| 3,177nt | | C → T | 28 | 2.34 |
| 3,373nt | | C → A | 43 | 3.59 |
| 6,874nt | | T → G | 72 | 6.01 |
| 6,883nt | | C → T | 38 | 3.17 |
| 8,887nt | nsp4 | A → G | 108 | 9.01 |
| 11,029nt | nsp6 | G → A | 41 | 3.42 |
| 11,083nt | | G → T | 32 | 2.67 |
| 12,316nt | nsp8 | A → G | 28 | 2.34 |
| 12,886nt | nsp9 | A → G | 39 | 3.25 |
| 13,599nt | RNA-dependent RNA polymerase | T → C | 63 | 5.25 |
| 14,408nt | | C → T | 1,180 | 98.42 |
| 15,540nt | | C → T | 29 | 2.42 |
| 19,839nt | endoRNAse | T → C | 105 | 8.76 |
| 20,268nt | | A → G | 47 | 3.92 |
| 21,724nt | Spike glycoprotein | G → A | 38 | 3.17 |
| 21,772nt | | C → T | 41 | 3.42 |
| 22,020nt | | T → C | 73 | 6.09 |
| 23,403nt | | A → G | 1,195 | 99.67 |

**Table 8** (continued)

| (B) | | | 01/19/2020–01/20/2021 | |
| --- | --- | --- | --- | --- |
| Position | Location | Mutation | Total Count | Percentage |
| 25,563nt | ORF3a | G → T | 43 | 3.59 |
| 26,750nt | Membrane glycoprotein | C → T | 53 | 4.42 |
| 27,415nt | ORF7a | G → T | 34 | 2.84 |
| 28,253nt | ORF8 | C → T | 32 | 2.67 |
| 28,881nt | Nucleocapsid phosphoprotein | G → A | 1,079 | 89.99 |
| 28,882nt | | G → A | 1,079 | 89.99 |
| 28,883nt | | G → C | 1,075 | 89.66 |
| 28,905nt | | C → T | 62 | 5.17 |
| 28,975nt | | G → T | 24 | 2 |
| 29,518nt | ORF10 | C → T | 49 | 4.09 |

The general design of these Tables follows the outline described in detail in the legend to Table 5 (United States). The number of sequences investigated for SARS-CoV-2 mutations is detailed in Tables for individual countries.

## Impact on coding capacity

The change in coding capacity of the long-term prevalent mutations in positions 241, 3,037, 14,408, and 23,403 was described for the US samples. Among the nine India-unique mutations, the following four led to functionally significant amino acid exchanges: position 2,292 (nsp2) gln–pro; 18,568 (3'–5'-exonuclease) leu–phe; 19,154 (3'–5'-exonuclease) thr–ile; and 28,311 (nucleocapsid phosphoprotein) ser–leu. Among the nine additional mutations, which were shared by one or several countries, only the following four led to amino acid exchanges: 6,312 (nsp3) thr–lys; 11,083 (nsp6) leu/tyr–phe; 21,724 (spike protein) leu-phe–phe-phe; 28,854 (nucleocapsid phosphoprotein) ser–leu (Table 6A). Again, many of the new SARS-CoV-2 mutations were responsible for functionally important non-synonymous amino acid exchanges in the corresponding protein.

## Brazil

In the nine SARS-CoV-2 mutations identified in a subset of about 100 published sequences available from Brazil in one time frame between 02/25 and 08/15, 2020 (Table 7A), five belonged to the worldwide prevalent hotspots at nucleotide numbers 241, 3,037, 14,408, 23,403, and 28,881. Two mutations at positions 12,053 and 25,088 were unique to the sequences from Brazil and were noted in between 15.7 and 34.4% of the analyzed sequences, respectively. Two of the novel shared mutations were also identified in sequences from France and Russia (27,299 and 29,148) at frequencies of about 40%. The mutation at nucleotide position 28,881 was found in 71.6% of the viral sequences studied. This mutation occurred in viral sequences from all countries investigated, except in those from China.

Of note, among the nine different new mutations observed in the SARS-CoV-2 isolates from Brazil, two were not observed in isolates from any of the eight other countries investigated. Possibly, they had recently emerged in the Brazilian population in which the virus had been replicating very actively, and the mutations had been selected under conditions of pandemic viral abundance. The frequent C → T mutations amounted to 44.4% frequency in this selection. The cutoff for temporal analysis was chosen before the variant strains P.1 and P.2 were identified. Table 7B presents the number and nature of individual mutations for all complete sequences with known sampling dates deposited to GISAID by January 20, 2021.

## Impact on coding capacity

The two Brazil-unique mutations at positions 12,053 (viral replicase) and 25,088 (viral spike protein) led to leu to phe and val to phe synonymous replacements, respectively. The two novel shared mutations at positions 27,299 (ORF6 protein) and 29,148 (nucleocapsid phosphoprotein) both caused ile to thr replacements of a non-synonymous nature.

Table 7B shows 27 individual mutations for the > 1,100 complete sequences with known sampling dates deposited to GISAID by January 20, 2021. The predominant mutations at positions 241, 3,037, 14,408, 23,403 showed frequencies at 99%. The mutation in the nucleocapsid phosphoprotein at position 28,881/2/3 presented with 93%, the highest frequency for this mutation among all 10 countries studied. As shown in Table 7A, in the time course study the nucleocapsid mutation reached a value of 71.6%. As of March 31, 71 cases of the B.1.1.7 variant from the UK and 641 cases of the P.1 variant (Table 2) were reported.

## Russia

Among the RU-I subsample of 226 SARS-CoV-2 RNA sequences analyzed between 03/24 and 06/07/2020 in the isolates from Russia, there were ten mutations of which six belonged to the previously described long-term prevalent mutations at positions 241, 3,037, 14,408, 23,403, 25,563, and 28,881 (Table 8A). The latter mutation in position 28,881 at a frequency of representation of 76.1% stood out in that it was not a point mutation but involved a three-nucleotide exchange creating a highly basic domain in the 3' terminal region of the SARS-CoV-2 nucleocapsid phosphoprotein as reported earlier (Weber et al, 2020). The four new mutations were located at sequence positions 3,140 (CC → TC, with a pro to asn-leu exchange in the amino acid sequence of nsp3, 20,268 (AG → GG, without change in amino acid composition in the endo RNase), 26,750 (CA →

**Table 9. France.**

| (A) | | | 04 – 09/12/2020 | |
|---|---|---|---|---|
| Position | Location | Mutation | Count | Incidence |
| 241nt | 5´UTR | CG → TG | 116/116 | prevalent |
| | | noneffective | | |
| 1,059nt | nsp2 | CC → TC | 16/116 | prevalent |
| | | A**CC** (Threonine) → A**TC** (Isoleucine) | | |
| 2,416nt | | CA → TA | 25/116 | CN,ES,RU,US, ZA |
| | | noneffective | | |
| 3,037nt | nsp3 | CT → TT | 115/116 | prevalent |
| | | noneffective | | |
| 4,543nt | | CA → TA | 15/116 | DE,ES |
| | | **CA**C (Histidine) → **TA**C (Tyrosine) | | |
| 5,629nt | | GT → TT | 15/116 | DE,ES |
| | | noneffective | | |
| 8,371nt | | GG → TG | 23/116 | ES,RU |
| | | CA**GG**TA (Glutamine Valine) → CA**TG**TA (Histidine Valine) | | |
| 9,526nt | nsp4 | GT → TT | 15/116 | DE,ES |
| | | AT**GT**CA (Methionine Serine) → AT**TT**CA (Isoleucine Serine) | | |
| 11,497nt | nsp6 | CT → TT | 15/116 | DE,ES |
| | | noneffective | | |
| 13,993nt | RNA-dependent RNA polymerase | GC → TC | 15/116 | DE,ES |
| | | **GC**T (Alanine) → **TC**T (Serine) | | |
| 14,408nt | | CT → TT | 114/116 | prevalent |
| | | C**CT** (Proline) → C**TT** (Leucine) | | |
| 15,324nt | | CA → TA | 22/116 | BR,CN,IN |
| | | noneffective | | |
| 15,766nt | | GT → TT | 15/116 | DE,ES |
| | | **GT**G (Valine) → **TT**G (Leucine) | | |
| 16,889nt | Helicase | AA → GA | 15/116 | DE,ES |
| | | A**AA** (Lysine) → A**GA** (Arginine) | | |
| 17,019nt | | GT → TT | 15/116 | DE,ES |
| | | GA**GT**TT (Glutamic Acid Phenylalanine) → GA**TT**TT (Aspartic Acid Phenylalanine) | | |
| 20,268nt | endoRNAse | AG → GG | 13/116 | ES,RU,US, ZA |
| | | noneffective | | |
| 22,992nt | Spike glycoprotein | GC → AC | 15/116 | DE,US |
| | | A**GC** (Serine) → A**AC** (Asparagine) | | |
| 23,403nt | | AT → GT | 116/116 | prevalent |
| | | G**AT** (Aspartic Acid) → G**GT** (Glycine) | | |
| 25,563nt | ORF3a protein | GA → TA | 57/116 | prevalent |
| | | CA**GA**GC (Glutamine Serine) → CA**TA**GC (Histidine Serine) | | |
| 25,710nt | | CT → TT | 16/116 | DE,ES |
| | | noneffective | | |
| 26,735nt | Membrane glycoprotein | CA → TA | 15/116 | DE,ES,IN, US |
| | | noneffective | | |

**Table 9** (continued)

| (A) | | | 04 – 09/12/2020 | |
|---|---|---|---|---|
| **Position** | **Location** | **Mutation** | **Count** | **Incidence** |
| 26,876nt | | TC → CC | 15/116 | DE,ES |
| | | noneffective | | |
| 28,833nt | Nucleocapsid phosphoprotein | CA → TA | 12/116 | ES |
| | | T**CA** (Serine) → T**TA** (Leucine) | | |
| 28,851nt | | GT → TT | 10/116 | IN |
| | | A**GT** (Serine) → A**TT** (Isoleucine) | | |
| 28,881nt | | GGG → AAC | 17/116 | prevalent |
| | | A**GGG**GA (Arginine Glycine) → A**AAC**GA (Lysine Arginine) | | |
| 28,975nt | | GT → CT | 15/116 | DE,ES,IN |
| | | AT**GT**CT (Methionine Serine) → AT**CT**CT (Isoleucine Serine) | | |
| 29,399nt | | GC → AC | 15/116 | DE,ES |
| | | **GC**T (Alanine) → **AC**T (Threonine) | | |

| (B) | | | 01/19/2020–01/20/2021 | |
|---|---|---|---|---|
| **Position** | **Location** | **Mutation** | **Total Count** | **Percentage** |
| 222nt | 5´UTR | C → T | 100 | 3.77 |
| 241nt | | C → T | 2,600 | 98 |
| 313nt | ORF1ab polyprotein → leader protein | C → T | 55 | 2.07 |
| 445nt | | T → C | 163 | 6.14 |
| 1,059nt | nsp2 | C → T | 385 | 14.51 |
| 2,416nt | | C → T | 320 | 12.06 |
| 3,037nt | nsp3 | C → T | 2,606 | 98.23 |
| 3,099nt | | C → T | 69 | 2.6 |
| 4,543nt | | C → T | 666 | 25.1 |
| 4,960nt | | G → T | 69 | 2.6 |
| 4,965nt | | C → T | 69 | 2.6 |
| 5,170nt | | C → T | 53 | 2 |
| 5,629nt | | G → T | 666 | 25.1 |
| 6,070nt | | C → T | 70 | 2.64 |
| 6,286nt | | C → T | 168 | 6.33 |
| 7,303nt | | C → T | 70 | 2.64 |
| 7,564nt | | C → T | 71 | 2.68 |
| 8,371nt | | G → T | 233 | 8.78 |
| 9,246nt | nsp4 | C → T | 69 | 2.6 |
| 9,526nt | | G → T | 667 | 25.14 |
| 10,279nt | 3C-like proteinase | C → T | 70 | 2.64 |
| 10,301nt | | C → A | 69 | 2.6 |
| 10,525nt | | C → T | 70 | 2.64 |
| 10,582nt | | C → T | 113 | 4.26 |
| 10,688nt | | G → T | 69 | 2.6 |
| 11,083nt | nsp6 | G → T | 99 | 3.73 |
| 11,132nt | | G → T | 54 | 2.04 |
| 11,497nt | | C → T | 666 | 25.1 |
| 11,851nt | nsp7 | G → T | 96 | 3.62 |

**Table 9** (continued)

| (B) | | | 01/19/2020–01/20/2021 | |
|---|---|---|---|---|
| **Position** | **Location** | **Mutation** | **Total Count** | **Percentage** |
| 13,993nt | RNA-dependent RNA polymerase | G → T | 664 | 25.03 |
| 14,230nt | | C → A | 68 | 2.56 |
| 14,408nt | | C → T | 2,606 | 98.23 |
| 15,324nt | | C → T | 467 | 17.6 |
| 15,738nt | | C → T | 63 | 2.37 |
| 15,766nt | | G → T | 667 | 25.14 |
| 16,889nt | Helicase | A → G | 665 | 25.07 |
| 17,019nt | | G → T | 665 | 25.07 |
| 18,877nt | 3'-to-5' exonuclease | C → T | 675 | 25.44 |
| 20,268nt | endoRNAse | A → G | 111 | 4.18 |
| 21,255nt | 2'-O-ribose methyltransferase | G → C | 167 | 6.29 |
| 21,800nt | Spike glycoprotein | G → T | 72 | 2.71 |
| 22,227nt | | C → T | 172 | 6.48 |
| 22,992nt | | G → A | 666 | 25.1 |
| 23,403nt | | A → G | 2,607 | 98.27 |
| 25,563nt | ORF3a | G → T | 1,474 | 55.56 |
| 25,688nt | | C → T | 56 | 2.11 |
| 25,710nt | | C → T | 677 | 25.52 |
| 26,735nt | Membrane glycoprotein | C → T | 670 | 25.25 |
| 26,801nt | | C → G | 167 | 6.29 |
| 26,876nt | | T → C | 667 | 25.14 |
| 27,632nt | ORF7a | G → T | 68 | 2.56 |
| 27,804nt | ORF7b | C → T | 85 | 3.2 |
| 28,830nt | Nucleocapsid phosphoprotein | C → A | 85 | 3.2 |
| 28,833nt | | C → T | 62 | 2.34 |
| 28,881nt | | G → A | 280 | 10.55 |
| 28,882nt | | G → A | 277 | 10.44 |
| 28,883nt | | G → C | 276 | 10.4 |
| 28,932nt | | C → T | 167 | 6.29 |
| 28,975nt | | G → C | 664 | 25.03 |
| 29,399nt | | G → A | 662 | 24.95 |
| 29,402nt | | G → T | 73 | 2.75 |
| 29,645nt | ORF10 | G → T | 169 | 6.37 |
| 29,779nt | 3′UTR | G → T | 67 | 2.53 |

The general design of these Tables follows the outline described in detail in the legend to Table 5 (United States). The number of sequences investigated for SARS-CoV-2 mutations is detailed in Tables for individual countries.

TA, without effect on the membrane glycoprotein), and at 27,415 (GC → TC, and an ala to ser change in the ORF6 protein).

Table 8B presents similar results of analyses on about 1,200 sequences collected during one year between 01/19/2020 and 01/20/2021. Again the prevalent mutations had reached close to 100% frequency, the nucleocapsid phosphoprotein about 90%. New mutations were not apparent. C to U transitions stood at 38% (Table 8B).

As of March 31, 2021, the detection of low numbers of variants B.1.1.7 and B.1.351 was reported from Russia.

**France**

Mutation frequencies were determined between 04 and 09/12, 2020 in 116 SARS-CoV-2 sequences, and a total of 27 mutations were documented. Among them, seven of the previously described long-term prevalent mutations were identified at frequencies as follows: nucleotide position 241 (100%), 1,059 (13.8%), 3,037 (99.1%), 14,408 (98.3%), 23,403 (100%), 25,563 (49.1%), 28,881 (14.7%). There were 20 new mutations at frequencies between 10 and 20%

**Table 10. Spain.**

| (A) | | | 06/01–09/20/2020 | |
|---|---|---|---|---|
| Position | Location | Mutation | Count | Incidence |
| 241nt | 5′UTR | CG → TG | 133/135 | prevalent |
| | | noneffective | | |
| 445nt | ORF1ab polyprotein → leader protein | TT → CT | 88/135 | CN,DE,FR |
| | | noneffective | | |
| 3,037nt | nsp3 | CT → TT | 131/135 | prevalent |
| | | noneffective | | |
| 5,572nt | | GT → TT | 11/135 | unique |
| | | AT**GT**AC (Methionine Tyrosine) → AT**TT**AC (Isoleucine Tyrosine) | | |
| 5,784nt | | CT → TT | 13/135 | unique |
| | | A**C**T (Threonine) → A**T**T (Isoleucine) | | |
| 6,286nt | | CT → TT | 89/135 | DE,FR,ZA |
| | | noneffective | | |
| 14,408nt | RNA-dependent RNA polymerase | CT → TT | 132/135 | prevalent |
| | | C**C**T (Proline) → C**T**T (Leucine) | | |
| 20,268nt | endoRNAse | AG → GG | 26/135 | FR,RU,US,ZA |
| | | noneffective | | |
| 21,255nt | 2′-O-ribose methyltransferase | GT → CT | 84/135 | DE,FR |
| | | noneffective | | |
| 22,227nt | Spike glycoprotein | CT → TT | 89/135 | DE,FR,ZA |
| | | noneffective | | |
| 22,297nt | | TA → CA | 11/135 | RU |
| | | noneffective | | |
| 25,049nt | | GA → TA | 18/135 | DE |
| | | **GA**T (Aspartic Acid) → **TA**T (Tyrosine) | | |
| 25,062nt | | GT → TT | 18/135 | unique |
| | | G**GT** (Glycine) → G**TT** (Valine) | | |
| 26,801nt | Membrane glycoprotein | CA → GA | 89/135 | DE,FR,ZA |
| | | noneffective | | |
| 27,944nt | ORF8 protein | CC → TC | 56/135 | FR |
| | | noneffective | | |
| 27,982nt | | CA → TA | 13/135 | unique |
| | | C**CA** (Proline) → C**TA** (Leucine) | | |
| 28,657nt | Nucleocapsid phosphoprotein | CG → TG | 19/135 | unique |
| | | noneffective | | |
| 28,881nt | | GGG → AAC | 14/135 | prevalent |
| | | A**GGG**GA (Arginine Glycine) → A**AAC**GA (Lysine Arginine) | | |
| 28,932nt | | CT → TT | 89/135 | unique |
| | | G**CT** (Alanine) → G**TT** (Valine) | | |
| 29,645nt | ORF10 protein | GT → TT | 89/135 | DE,FR |
| | | noneffective | | |

**Table 10** (continued)

| (B) | | | 01/19/2020–01/20/2021 | |
|---|---|---|---|---|
| Position | Location | Mutation | Total Count | Percentage |
| 241nt | 5′UTR | C → T | 2,690 | 78.47 |
| 313nt | ORF1ab polyprotein → leader protein | C → T | 117 | 3.41 |
| 445nt | | T → C | 858 | 25.03 |
| 1,059nt | nsp2 | C → T | 122 | 3.56 |
| 1,987nt | | A → G | 75 | 2.19 |
| 3,037nt | nsp3 | C → T | 2,717 | 79.26 |
| 5,170nt | | C → T | 141 | 4.11 |
| 6,286nt | | C → T | 861 | 25.12 |
| 6,294nt | | T → C | 82 | 2.39 |
| 8,782nt | nsp4 | C → T | 601 | 17.53 |
| 9,477nt | | T → A | 379 | 11.06 |
| 11,083nt | nsp6 | G → T | 166 | 4.84 |
| 11,132nt | | G → T | 137 | 4 |
| 13,006nt | nsp9 | T → C | 77 | 2.25 |
| 14,408nt | RNA-dependent RNA polymerase | C → T | 2,708 | 79 |
| 14,805nt | | C → T | 408 | 11.9 |
| 20,268nt | endoRNAse | A → G | 1,223 | 35.68 |
| 21,255nt | 2′-O-ribose methyltransferase | G → C | 780 | 22.75 |
| 22,227nt | Spike glycoprotein | C → T | 843 | 24.59 |
| 23,403nt | | A → G | 2,731 | 79.67 |
| 25,049nt | | G → T | 71 | 2.07 |
| 25,563nt | ORF3a | G → T | 147 | 4.29 |
| 25,688nt | | C → T | 78 | 2.28 |
| 25,979nt | | G → T | 371 | 10.82 |
| 26,088nt | | C → T | 215 | 6.27 |
| 26,144nt | | G → T | 100 | 2.92 |
| 26,801nt | Membrane glycoprotein | C → G | 855 | 24.94 |
| 27,944nt | ORF8 | C → T | 456 | 13.3 |
| 28,144nt | | T → C | 599 | 17.47 |
| 28,657nt | Nucleocapsid phosphoprotein | C → T | 441 | 12.86 |
| 28,863nt | | C → T | 378 | 11.03 |
| 28,881nt | | G → A | 398 | 11.61 |
| 28,882nt | | G → A | 396 | 11.55 |
| 28,883nt | | G → C | 395 | 11.52 |
| 28,932nt | | C → T | 850 | 24.8 |
| 29,645nt | ORF10 | G → T | 840 | 24.5 |
| 29,734nt | 3′UTR | G → C | 302 | 8.81 |
| 29,870nt | | C → A | 107 | 3.12 |

The general design of these Tables follows the outline described in detail in the legend to Table 5 (United States). The number of sequences investigated for SARS-CoV-2 mutations is detailed in Tables for individual countries.

that were not described previously (Weber *et al*, 2020). C-U transitions reached 40.7% (Tables 9A and 13). Of interest, none of the new mutations was unique to France in the 116 sequences displayed in Table 9A. Instead, a large percentage of the mutations were shared with Germany and Spain, both neighboring countries. Most novel mutations occurred at frequencies between 10 and 20% (Table 9A). Among the novel mutations, 20 occurred at > 10, many of them > 20% frequencies.

**Table 11. Germany.**

| (A) Position | Location | Mutation | 02–03/23 2020* Count | 02–06/ 17/2020 Count | 06/24–08/ 28/2020 Count | 09/10–10/ 13/2020 Count | Incidence |
|---|---|---|---|---|---|---|---|
| 241nt | 5′UTR | CG → TG | 4/62 | 112/138 | 17/17 | 70/70 | Prevalent |
| | | noneffective | | | | | |
| 445nt | nsp1 | TT → CT | 0/62 | 0/138 | 1/17 | 17/70 | CN,FR |
| | | **TT**G (Leucine) → GT**CT**TG (Valine Leucine) | | | | | |
| 1,059nt | nsp2 | CC → TC | 21/62 | 27/138 | 0/17 | 2/70 | Prevalent |
| | | A**CC** (Threonine) → A**TC** (Isoleucine) | | | | | |
| 1,440nt | | GC → AC | 15/62 | 18/138 | 0/17 | 0/70 | US |
| | | G**GC** (Glycine) → G**AC** (Aspartic Acid) | | | | | |
| 1,513nt | | CC → TC | 0/62 | 0/138 | 0/17 | 13/70 | Unique |
| | | noneffective | | | | | |
| 2,891nt | | GC → AC | 15/62 | 18/138 | 0/17 | 0/70 | US |
| | | **GC**A (Alanine) → **AC**A (Threonine) | | | | | |
| 3,037nt | nsp3 | CT → TT | 41/62 | 114/138 | 17/17 | 70/70 | Prevalent |
| | | noneffective | | | | | |
| 3,602nt | | CA → TA | 0/62 | 0/138 | 5/17 | 6/70 | Unique |
| | | **CA**C (Histidine) → **TA**C (Tyrosine) | | | | | |
| 4,543nt | | CA → TA | 0/62 | 0/138 | 5/17 | 2/70 | ES,FR,US |
| | | noneffective | | | | | |
| 6,286nt | | CT → TT | 0/62 | 0/138 | 1/17 | 17/70 | ES,FR,ZA |
| | | noneffective | | | | | |
| 6,941nt | | CT → TT | 0/62 | 0/138 | 5/17 | 6/70 | Unique |
| | | noneffective | | | | | |
| 14,408nt | RNA-dependent RNA polymerase | CT → TT | 39/62 | 114/138 | 17/17 | 70/70 | Prevalent |
| | | C**CT** (Proline) → C**TT** (Leucine) | | | | | |
| 15,324nt | | CA → TA | 1/62 | 1/138 | 5/17 | 6/70 | BR,CN,FR, IN |
| | | noneffective | | | | | |
| 16,075nt | | GA → TA | 0/62 | 0/138 | 0/17 | 11/70 | FR |
| | | G**AT** (Aspartic Acid) → **TA**T (Tyrosine) | | | | | |
| 19,839nt | endoRNAse | TA → CA | 0/62 | 0/138 | 2/17 | 11/70 | CN,ES,FR, IN,US |
| | | noneffective | | | | | |
| 21,255nt | 2′-O-ribose methyltransferase | GT → CT | 0/62 | 0/138 | 1/17 | 17/70 | ES,FR |
| | | noneffective | | | | | |
| 21,855nt | Spike glycoprotein | CT → TT | 0/62 | 0/138 | 5/17 | 6/70 | ZA |
| | | T**CT** (Serine) → T**TT** (Phenylalanine) | | | | | |
| 22,227nt | | CT → TT | 0/62 | 0/138 | 1/17 | 18/70 | ES,FR,ZA |
| | | noneffective | | | | | |
| 22,346nt | | GC → TC | 0/62 | 0/138 | 0/17 | 13/70 | Unique |
| | | **GC**T (Alanine) → **TC**T (Serine) | | | | | |
| 22,377nt | | CT → TT | 0/62 | 0/138 | 0/17 | 13/70 | Unique |
| | | C**CT** (Proline) → C**TT** (Leucine) | | | | | |
| 23,403nt | | AT → GT | 1/62 | 112/138 | 17/17 | 70/70 | Prevalent |
| | | G**AT** (Aspartic Acid) → G**GT** (Glycine) | | | | | |

**Table 11** (continued)

| (A) Position | Location | Mutation | 02–03/23 2020* Count | 02–06/17/2020 Count | 06/24–08/28/2020 Count | 09/10–10/13/2020 Count | Incidence |
|---|---|---|---|---|---|---|---|
| 25,505nt | ORF3a protein | AA → GA | 0/62 | 0/138 | 5/17 | 6/70 | Unique |
| | | C**AA** (Glutamine) → C**GA** (Arginine) | | | | | |
| 25,563nt | | GA → TA | 21/62 | 27/138 | 2/17 | 5/70 | Prevalent |
| | | CA**GA**GC (Glutamine Serine) → CA**TA**GC (Histidine Serine) | | | | | |
| 25,906nt | | GG → CG | 0/62 | 0/138 | 5/17 | 6/70 | Unique |
| | | **GG**T (Glycine) → **CG**T (Arginine) | | | | | |
| 26,801nt | Membrane glycoprotein | CA → GA | 1/62 | 0/138 | 1/17 | 17/70 | ES,FR,ZA |
| | | noneffective | | | | | |
| 27,046nt | | CG → TG | 1/62 | 16/138 | 3/17 | 0/70 | BR,RU |
| | | A**CG** (Threonine) → A**TG** (Methionine) | | | | | |
| 28,651nt | Nucleocapsid phosphoprotein | CA → TA | 0/62 | 0/138 | 5/17 | 6/70 | FR,RU |
| | | noneffective | | | | | |
| 28,706nt | | CA → TA | 0/62 | 0/138 | 0/17 | 11/70 | Unique |
| | | **CA**C (Histidine) → **TA**C (Tyrosine) | | | | | |
| 28,869nt | | CA → TA | 0/62 | 0/138 | 5/17 | 6/70 | Unique |
| | | C**CA** (Proline) → C**TA** (Leucine) | | | | | |
| 28,881nt | | GGG → AAC | 9/62 | 35/138 | 9/17 | 38/70 | Prevalent |
| | | A**GGG**GA (Arginine Glycine) → A**AAC**GA (Lysine Arginine) | | | | | |
| 28,932nt | | CT → TT | 0/62 | 0/138 | 1/17 | 17/70 | FR |
| | | G**CT** (Alanine) → G**TT** (Valine) | | | | | |
| 29,645nt | ORF10 protein | GT → TT | 0/62 | 0/138 | 1/17 | 17/70 | ES,FR |
| | | noneffective | | | | | |
| 29,751nt | 3′UTR | GA → CA | 0/62 | 0/138 | 0/17 | 11/70 | Unique |
| | | noneffective | | | | | |

| (B) Position | Location | Mutation | 01/19/2020–01/20/2021 Total count | Percentage |
|---|---|---|---|---|
| 187 nt | 5′UTR | A→G | 45 | 2.18 |
| 204 nt | | G→T | 48 | 2.32 |
| 241 nt | | C→T | 1,790 | 86.64 |
| 313 nt | ORF1ab polyprotein → leader protein | C→T | 53 | 2.57 |
| 445 nt | | T→C | 159 | 7.7 |
| 1,059 nt | nsp2 | C→T | 399 | 19.31 |
| 1,440 nt | | G→A | 76 | 3.68 |
| 2,891 nt | nsp3 | G→A | 76 | 3.68 |
| 3,037 nt | | C→T | 1,796 | 86.93 |
| 3,373 nt | | C→A | 53 | 2.57 |
| 3,602 nt | | C→T | 77 | 3.73 |
| 4,543 nt | | C→T | 42 | 2.03 |
| 6,286 nt | | C→T | 155 | 7.5 |
| 6,406 nt | | C→T | 57 | 2.76 |
| 6,941 nt | | C→T | 79 | 3.82 |

**Table 11** (continued)

| (B) | | | 01/19/2020–01/20/2021 | |
|---|---|---|---|---|
| Position | Location | Mutation | Total count | Percentage |
| 8,782 nt | nsp4 | C→T | 128 | 6.2 |
| 11,083 nt | nsp6 | G→T | 91 | 4.4 |
| 14,408 nt | RNA-dependent RNA polymerase | C→T | 1,782 | 86.25 |
| 14,805 nt | | C→T | 49 | 2.37 |
| 15,324 nt | | C→T | 138 | 6.68 |
| 18,877 nt | 3′-to-5′ exonuclease | C→T | 55 | 2.66 |
| 18,972 nt | | G→A | 58 | 2.81 |
| 19,839 nt | endoRNAse | T→C | 52 | 2.52 |
| 20,268 nt | | A→G | 78 | 3.78 |
| 21,255 nt | 2′-O-ribose methyltransferase | G→C | 162 | 7.84 |
| 21,614 nt | Spike glycoprotein | C→T | 45 | 2.18 |
| 21,855 nt | | C→T | 76 | 3.68 |
| 22,227 nt | | C→T | 166 | 8.03 |
| 22,468 nt | | G→T | 116 | 5.61 |
| 23,403 nt | | A→G | 1,800 | 87.12 |
| 25,505 nt | ORF3a | A→G | 74 | 3.58 |
| 25,550 nt | | T→A | 53 | 2.57 |
| 25,563 nt | | G→T | 492 | 23.81 |
| 25,906 nt | | G→C | 74 | 3.58 |
| 25,922 nt | | G→T | 50 | 2.42 |
| 25,996 nt | | G→T | 75 | 3.63 |
| 26,144 nt | | G→T | 44 | 2.13 |
| 26,530 nt | Membrane glycoprotein | A→G | 55 | 2.66 |
| 26,735 nt | | C→T | 43 | 2.08 |
| 26,801 nt | | C→G | 145 | 7.02 |
| 27,046 nt | | C→T | 68 | 3.29 |
| 27,944 nt | ORF8 | C→T | 89 | 4.31 |
| 28,144 nt | | T→C | 131 | 6.34 |
| 28,651 nt | Nucleocapsid phosphoprotein | C→T | 74 | 3.58 |
| 28,854 nt | | C→T | 59 | 2.86 |
| 28,869 nt | | C→T | 75 | 3.63 |
| 28,878 nt | | G→A | 124 | 6 |
| 28,881 nt | | G→A | 589 | 28.51 |
| 28,882 nt | | G→A | 585 | 28.32 |
| 28,883 nt | | G→C | 585 | 28.32 |
| 28,932 nt | | C→T | 162 | 7.84 |
| 29,645 nt | ORF10 | G→T | 161 | 7.79 |

The general design of these Tables follows the outline described in detail in the legend to Table 5 (United States). The number of sequences investigated for SARS-CoV-2 mutations is detailed in Tables for individual countries.

Table 9B lists mutational frequency in sequences deposited up until January 20, 2021. As of March 31, 2021, the frequencies of variants of concern, B.1.1.7, B.1.351, P1, B.1.429 + B.1.427, and B.1.525 were 6,290, 537, 38, 4, and 30, respectively (Table 2). As complete sequence analyses on COVID-19 isolates are progressing rapidly, new data on the emergence of new variants can be expected.

**Impact on coding capacity**

Among these 20 not-previously described novel mutations, eight did not affect the coding capacity of the relevant viral proteins. Most of the 12 coding-relevant mutations led to amino acid exchanges that were non-synonymous: nsp2, 3, 4, RNA-dependent RNA

**Table 12. China.**

| (A) Position | Location | Mutation | 12/23/2019–03/18/2020* Count | 03/20–07/22/2020 Count | Incidence |
|---|---|---|---|---|---|
| 241nt | 5′UTR | CG → TG<br>noneffective | 0/98 | 23/33 | Prevalent |
| 3,037nt | nsp3 | CT → TT<br>noneffective | 2/99 | 23/33 | Prevalent |
| 8,782nt | nsp4 | CC → TC<br>noneffective | 29/99 | 0/33 | DE,ES,IN,US |
| 14,408nt | RNA-dependent RNA polymerase | CT → TT<br>C**CT** (Proline) → C**TT** (Leucine) | 2/99 | 19/33 | Prevalent |
| 23,403nt | Spike glycoprotein | AT → GT<br>G**AT** (Aspartic Acid) → G**GT** (Glycine) | 2/99 | 22/33 | Prevalent |
| 28,144nt | ORF8 protein | TA → CA<br>T**TA** (Leucine) → T**CA** (Serine) | 29/99 | 0/33 | DE,ES,IN,US |
| 28,881nt | Nucleocapsid phosphoprotein | GGG → AAC<br>A**GGG**GA (Arginine Glycine) → A**AAC**GA (Lysine Arginine) | 2/99 | 11/33 | Prevalent |

| (B) Position | Location | Mutation | 01/19/2020–01/20/2021 Total Count | Percentage |
|---|---|---|---|---|
| 4nt | 5′UTR | A → G | 15 | 2.49 |
| 241nt | | C → T | 68 | 11.28 |
| 1,397nt | nsp2 | G → A | 18 | 2.99 |
| 2,392nt | | T → C | 13 | 2.16 |
| 3,037nt | nsp3 | C → T | 65 | 10.78 |
| 6,354nt | | C → T | 14 | 2.32 |
| 7,075nt | | T → C | 14 | 2.32 |
| 8,022nt | | T → G | 15 | 2.49 |
| 8,782nt | nsp4 | C → T | 191 | 31.67 |
| 10,747nt | 3C-like proteinase | C → T | 14 | 2.32 |
| 11,083nt | nsp6 | G → T | 40 | 6.63 |
| 11,794nt | | A → G | 14 | 2.32 |
| 14,408nt | RNA-dependent RNA polymerase | C → T | 55 | 9.12 |
| 15,324nt | | C → T | 13 | 2.16 |
| 15,342nt | | C → T | 14 | 2.32 |
| 15,360nt | | C → T | 14 | 2.32 |
| 15,666nt | | G → A | 14 | 2.32 |
| 16,733nt | Helicase | C → T | 14 | 2.32 |
| 17,373nt | | C → T | 26 | 4.31 |
| 18,060nt | 3'-to-5' exonuclease | C → T | 16 | 2.65 |
| 21,707nt | spike glycoprotein | C → T | 24 | 3.98 |
| 21,727nt | | C → T | 14 | 2.32 |
| 22,020nt | | T → C | 16 | 2.65 |
| 23,403nt | | A → G | 67 | 11.11 |
| 25,416nt | ORF3a | C → T | 14 | 2.32 |
| 26,144nt | | G → T | 39 | 6.47 |
| 27,213nt | ORF6 | C → T | 15 | 2.49 |

**Table 12** (continued)

| (B) Position | Location | Mutation | 01/19/2020–01/20/2021 Total Count | Percentage |
|---|---|---|---|---|
| 28,144nt | ORF8 | T → C | 212 | 35.16 |
| 28,688nt | Nucleocapsid phosphoprotein | T → C | 13 | 2.16 |
| 28,854nt | | C → T | 14 | 2.32 |
| 28,881nt | | G → A | 33 | 5.47 |
| 28,882nt | | G → A | 31 | 5.14 |
| 28,883nt | | G → C | 31 | 5.14 |
| 29,095nt | | C → T | 31 | 5.14 |
| 29,742nt | 3′UTR | G → T | 19 | 3.15 |
| 29,835nt | | C → T | 14 | 2.32 |

The general design of these Tables follows the outline described in detail in the legend to Table 5 (United States). The number of sequences investigated for SARS-CoV-2 mutations is detailed in Tables for individual countries.

polymerase, the helicase, the endoRNAse, the spike glycoprotein, and the nucleocapsid phosphoprotein (Table 9A and B).

**Spain**

During the period between 06/01 and 09/20/2020, we analyzed 135 sequences and observed 20 mutations in the Spanish isolates (Table 10A). Of these, four, the long-term prevalent ones, had been described earlier in positions 241, 3,037, 14,408, and 28,881. Except for the latter one at 10.4% frequency, the three former came close to 100% occurrence. Of the 16 new mutations, six occurred in Spanish isolates exclusively (termed unique), namely in positions 5,572

(GT → TT, frequency 8.1%, changing the amino acid sequence met to ile in nsp3), 5,784 (CT → TT, frequency 9.6%, thr to ile in nsp3), 25,062 (GT → TT, frequency 13.3%, amino acid change gly to val in the spike glycoprotein), 27,982 (CA → TA, frequency 9.6%, changing the sequence from pro to leu in the ORF8 protein), 28,657 (CG → TG, at frequency of 14.1%, without affecting the nucleocapsid phosphoprotein), and 28,932 (CT → TT at frequency of 65.9% and altering the amino acid composition in this position in the nucleocapsid phosphoprotein from ala to val). The remaining 10 novel shared mutants were also found in isolates from other countries and were located in positions as shown in previous tables. With the exception of a point mutation at position 25,049 in the spike

**Table 13. Survey.**

| Country | Total number mutations | Novel Unique mutations | Novel Shared mutations | Sum novel mutations | Prevalent mutations | C to T transitions [in % of mutants] | RNA replication | Spike glycoprotein | Nucleocapsid phosphoprotein | COVID-19 cases | COVID-19 deaths |
|---|---|---|---|---|---|---|---|---|---|---|---|
| United Kingdom | 43 | 20 | 18 | 38 (88.4%) | 5 | 53.6 | 8 | 6 | 4 | 3,617,459 | 97,329 (2.69%) |
| South Africa | 28 | 9 | 12 | 21 (75%) | 7 | 48.1 | 4 | 7 | 3 | 1,404,839 | 40,574 (2.89%) |
| United States | 39 | 17 | 13 | 30 (76.9%) | 7 | 61.5 | 13 | 3 | 7 | 25,546,140 | 427,294 (1.67%) |
| India | 23 | 9 | 9 | 18 (78.3%) | 5 | 65.2 | 6 | 4 | 2 | 10,655,435 | 153,376 (1.44%) |
| Brazil | 9 | 2 | 2 | 4 (44.4%) | 5 | 44.4 | 1 | 2 | 2 | 8,816,254 | 216,445 (2.46%) |
| Russia | 10 | 3 | 1 | 4 (40%) | 6 | 50 | 2 | 1 | 1 | 3,698,273 | 68,971 (1.86%) |
| France | 27 | 0 | 20 | 20 (74.1%) | 7 | 40.7 | 7 | 2 | 5 | 3,035,181 | 72,877 (2.40%) |
| Spain | 20 | 6 | 10 | 16 (80%) | 4 | 50 | 3 | 4 | 3 | 2,603,472 | 55,441 (2.13%) |
| Germany | 33 | 11 | 15 | 26 (78.8%) | 7 | 51.5 | 5 | 5 | 5 | 2,137,689 | 52,536 (2.46%) |
| People's Republic of China | 7 | 0 | 2 | 2 (28.6%) | 5 | 57.1 | 1 | 1 | 1 | 88,911 | 4,635 (5.21%) |

The rise of new SARS-CoV-2 mutations in many countries was juxtaposed to the high COVID-19 incidence values around the world. The mutants and their frequencies compiled and calculated in this Table were based on the data presented in Tables 3 and 4A to 12A. World incidence of COVID-19, as of January 30, 2021, in 219 countries was COVID-19 cases—102,87 million, fatalities—2.22 million (columns 10 and 11). Column 5 lists the total of novel mutations for each country, percentage values related this sum to the total number of mutations. Source for worldwide spread of COVID-19—https://www.worldometers.info/corona virus/.

The UK data in this Table do not contain results from the analysis of the SARS-CoV-2 variant B.1.1.7 which are shown in Table 1, as of April 01, 2021.

glycoprotein and an ensuing amino acid exchange from asp to tyr, none of the other nine mutations in the shared category led to an amino acid exchange. We also note that in the Spanish collection of SARS-CoV-2 mutations, there were four in the spike glycoprotein that were all different from the well-known position 23,403. Two of these new spike mutations led to non-synonymous amino acid exchanges in the spike glycoprotein: in position 25,049 asp to tyr and in 25,062 gly to val (Table 10A).

Non-synonymous mutations might become relevant when evaluating the efficacy of a solely spike-directed SARS-CoV-2 vaccine. As a note of caution, one should not rule out functional consequences of nominally silent mutations for SARS-CoV-2 competence, since they might affect the secondary structure of the viral RNA with sequelae in replication and relevant interactions of the viral genome with viral and/or cellular proteins. Moreover, more far reaching consequences of SARS-CoV-2 mutations like their effects on translation efficiency or codon choice might become important when trying to understand differences in viral transmissibility and pathogenesis.

It is interesting to note that, although the latest Spanish collection of SARS-CoV-2 mutations contains four mutations in the spike glycoprotein, at earlier time points the D614G mutation in position 23,403 was not present (Table 10A). In Table 10B, describing mutant frequencies between 01/19/2020 and 01/20/2021, the 23,403 mutant was present at about 80%, whereas in France and England prevalence was > 96%. Moreover, for the 01/2020 to 01/2021 period, mutations in 38 sequences lay above the 2% cutoff. The predominant mutations reached values around 80% representation. C to U (T) transitions were at 50%. Among the novel mutations, 17 showed prevalence of > 10%, eight of them of > 20%.

As of March 31, 2021, the frequencies of variants of concern, B.1.1.7, B.1.351, P1, B.1.429 + B.1.427, and B.1.525 were 4,352, 31, 20, 2, and 18, respectively (Table 2).

## Germany

During the course of the pandemic, we tabulated the occurrence of SARS-CoV-2 mutants which arose between February to March 23 (DE-I) (Weber *et al*, 2020), February to June 17 (DE-II), June 24 to August 28 (DE-III), the latter isolates with only 17 sequences available for analyses, and September 10 to October 13 (DE-IV) with 70 sequences. Apart from the prevalent mutations, there were relatively few mutations exceeding 10% representation in the time frame of DE-II. Among the total of 33 mutations in the SARS-CoV-2 RNA sequence (Table 11A), seven belonged to the previously described collection of long-term prevalent sequences—at positions 241, 1,059, 3,037, 14,408, 23,403, 25,563, 28,881 with coding frame alterations as outlined in previous Tables. In the DE-III sample, four of these long-term prevalent mutations had reached 100% representation, two had disappeared, and the mutation at 28,881 had remained at about 53%. Six mutations could be detected exclusively in the DE-III samples from Germany, in positions 3,602 (CA → TA), 6,941 (CT → TT), 21,855 (CT → TT), 25,505 (AA → GA), 25,906 (GG → CG), 28,869 (CA → TA), all of them at 29% of representation. There were mutations in six positions which had been observed also in isolates from other countries, as indicated, and all of them showed modest frequencies. It is interesting to note that 52% of the mutations detected in sequences from France were shared with Germany, but only 16% of the mutations identified in

Germany were shared with those from France (Table 9A). During the time interval of about a month, September 10 to October 13 (DE-IV), that immediately preceded a marked rise in COVID-19 cases in Germany, 23 new mutations were identified six of which reached a prevalence of > 20% and seven of > 10% in the SARS-CoV-2 sequences studied. During the same period, 4 of the prevalent mutations were represented in 100% of sequences, one, at 28,881 in 54%.

Table 11B lists the total number of mutations and variants up until January 20, 2021, from GISAID complete sequences with 52 entries at > 2% incidence. The prevalent mutations reach about 86% occurrence. Only at three sites, mutations were found at > 10%. C to U transitions were recorded in 46% of the studied sites (Table 11B).

## Impact on coding capacity

With the exception of the point mutation at 6,941 which was synonymous, the five other mutations were non-synonymous: 3,602 his to tyr (nsp3); 21,855 ser to phe (nsp3); 25,505 glu to arg (ORF3a protein); 25,906 gly to arg (ORF3a protein); and 28,869 pro to leu (nucleocapsid phosphoprotein).

As of March 31, 2021, the frequencies of variants of concern, B.1.1.7, B.1.351, P1, B.1.429 + B.1.427, and B.1.525 were 21,038, 652, 63, 6, and 123, respectively (Table 2).

## China

In December of 2020, the first cases of COVID-19 emerged in Wuhan, Hubei Province in China, reportedly among workers and customers of the Huanan Seafood Market. The Chinese authorities eventually reacted with a very strict shutdown in Hubei Province, the epicenter of COVID-19, to limit the spread of the new disease. At present, most new cases of COVID-19 are reportedly being registered in Shanghai and a few additional places. The analyses of SARS-CoV-2 mutants up to March 18, 2020 (CN-I), revealed point mutations in only two genome positions, 8,782 (CC → TC, without amino acid exchanges) and 28,144 (TA → CA causing a leu to ser exchange in ORF8 protein), both at frequencies of 29.3% (Table 12A). An extension of our mutant research among a relatively limited number of published sequences to the period from March 20 to June 22, 2020 (CN-II), revealed mutations in five of the long-term prevalently affected sequence positions: 241 (CG → TG at a frequency of 69.7% without coding changes), 3,037 (CT → TT, at a frequency of 69.7%, without coding changes), 14,408 (CT → TT at a frequency of 57.6% and a codon change pro to leu in the gene for the RNA-dependent RNA polymerase), 23,403 (AT → GT at a frequency of 66.7% and an asp to gly exchange in the spike glycoprotein), and at 28,881 (GGG → AAC at a frequency of 33.3% and the codon exchange arg-gly to lys-arg, reported previously). Remarkably, the novel shared point mutations in positions 8,782 and 28,144 had disappeared at the later time point (Table 12). These latter mutations may have been introduced to China by visitors or business travelers and then died out because they did not confer a strong evolutionary advantage or due to not enough sequencing. The total counts of mutations up until January 20 are presented in Table 12B. There are only very scant data on the occurrence of variants from China (Table 2).

# Discussion

## SARS-CoV-2 genetics will require in-depth analyses

It has been the intent of this project to follow the genetic evolution of SARS-CoV-2 after the virus transgressed a host barrier and during the ensuing major pandemic in the human population. The virus has shown great replicative and mutagenic potential and penetrated into the large human population of 7.8 billion that lacked previous encounters with SARS-CoV-2. In this context, the primary question was not to understand viral mutagenesis in general in its biochemical or genetic details, but to identify mutants that showed the potential to become prevalent with possible fitness advantages. Which mutants and variants would have the capability to persist and multiply in the course of rapid spread of SARS-CoV-2 within the human population? It will be a continuing long-term challenge to pursue the outcome and time course of a competition in that 29,903 nucleotides in the viral genome were pitted against about 3 billion in the human genome. The SARS-CoV-2 has a repertoire of mutable sites in a stretch of 29,903 nucleotides that cannot only be varied by introducing point mutations but be extended by an almost inexhaustible combination of multiple mutations in the same genome, by deletions and insertions. Before the viral dominance in the human population began, SARS-CoV-2 had already made a major leap, its transition from an animal to the novel human host, an undocumented step in its own right in which mutagenesis and selection must have played a major role. Thus, the impact of ethnic and socio-economic differences in the human population will have to be considered as important factors. In a summary of all mutation analyses, we have compared the number and types of mutations to the extent of the COVID-19 pandemic in ten different countries that currently report high numbers of cases and fatalities (Table 13).

Of course, this summary offers only a broad temporal correlation of mutant data and extent of the pandemic in individual countries. High current incidence of COVID-19 is paralleled by high numbers of new mutations and variants, although this relationship was not observed in Brazil or Russia, possibly because the relevant data from these countries have not been available. In anticipation, it will be a further challenge to evaluate the real-world success of the numerous COVID-19 vaccination programs.

More than 650 publications on the "evolution of SARS-CoV-2 genomes" have been listed under PubMed which is evidence for the sustained interest in this research topic. Here, we cannot meaningfully summarize this extensive literature. A recent publication (MacLean *et al,* 2021) investigated how natural selection in the likely original host of SARS-CoV-2, *Sarbecoviruses* in horseshoe bats, might have facilitated the rise of a "generalist virus" that presumably, without major further mutagenesis, had become fully equipped to function as an efficient human to human pathogen. The authors conceded the possible existence of an intermediate host between bats and humans. In fact, the origin of SARS-CoV-2 remains unclear and will remain an area of continuing investigations and debate [WHO-2019-nCoV-FAQ-Virus_origin-2020.1-eng.pdf (122.6K)].

## Replication and selection

Rapid worldwide replication of SARS-CoV-2 in heterogeneous populations has been paralleled by the rise of novel mutations. In this report, we have studied mutations in SARS-CoV-2 RNA sequences isolated in the UK, South Africa, Brazil, the United States, India, Russia, France, Spain, Germany, and China that have become available in the GISAID database during a one-year period between January 19, 2020, and January 20, 2021, and beyond to March 31, 2021 (revised Tables 1 and 2). We have examined the rise of novel mutations both using sequence subsets segregated by date and also overall in a large cross-section. It seems that throughout 2020 and into the first quarter of 2021, more mutations in combination were found and propagated rapidly despite lockdowns and other efforts to contain the spread, perhaps owing to potential increased transmissibility and pathogenicity of SARS-CoV-2. The current data are compatible with the interpretation that rapid regional expansion and efficient viral replication in human populations of very different genetic and socio-economic backgrounds enhance the selection of new mutations in the viral RNA genome. Differences in defense mechanisms operative in various populations infected by SARS-CoV-2 and/or the various therapeutic measures employed in fighting the infection might also have influenced the selection of new mutants. It is uncertain whether there was region-specific selection of specific mutations or whether other factors might have furthered differences in unique versus shared novel mutations.

Figure 1 and Tables 1 and 2 document the number of novel variants in each country as of March 31, 2021. The speed by which the virus travelled even during lockdowns emphasizes the difficulty in suppressing transmission of highly contagious respiratory viruses. By now, it has become apparent that the new variants can be associated with increased pathogenesis although more research needs to be done. The preliminary finding of increased transmissibility of the B.1.1.7 and B.135 variant hinders efforts to contain the virus (https://khub.net/documents/135939561/338928724/SARS-CoV-2 + variant + under + investigation%2C + meeting + minutes.pdf/962e866b-161f-2fd5-1030-32b6ab467896; https://virological.org/t/preliminary-genomic-characterisation-of-an-emergent-sars-cov-2-lineage-in-the-uk-defined-by-a-novel-set-of-spike-mutations/563; Rambaut *et al,* 2020; preprint: Cheng *et al,* 2021; https://www.cogconsortium.uk/wp-content/uploads/2021/01/Report-2_COG-UK_SARS-CoV-2-Mutations.pdf; Tegally *et al,* 2021; Volz *et al,* 2021; Wibmer *et al,* 2021). The vaccines are expected to work against the novel variants, although with some at reduced efficacy (https://www.janssen.com/johnson-johnson-announces-single-shot-janssen-covid-19-vaccine-candidate-metprimary-endpoints); https://ir.novavax.com/news-releases/news-release-details/novavax-covid-19-vaccine-demonstrates-893-efficacy-uk-phase-3; preprint: Wang *et al,* 2021; Xie *et al,* 2021), but caution is urged to watch more aggressive viral evolution as a consequence of vaccination programs.

## Rise of novel mutations and variants with new properties—A hypothesis

After initially demonstrating the prevalence of about 10 mutants in at least 10 different countries, SARS-CoV-2 evolved to display new point mutations worldwide that were selected among affected populations in a period of weeks (Tables 3, 4A, B to 12A, B). As shown in Table 13, column 5, the number of novel point mutations in some of the countries analyzed ranged between 16 and 38. As a consequence of highly efficient sequencing programs in the UK (UK

Consort), previously not recognized variants have started to appear in late 2020 and are currently spreading worldwide (Figure 1, Table 2). The impact of these and future variants on potential increases in viral pathogenicity cannot be predicted at present. There is recent evidence that the B.1.1.7 variant has shown increased infectivity of SARS-CoV-2 as well as more severe forms and higher mortality of COVID-19 (Davies *et al*, 2021; https://khub. net/documents/135939561/338928724/SARS-CoV-2 + variant + under + investigation%2C + meeting + minutes.pdf/962e866b-161f-2fd5-1030-32b6ab467896).

The incidence of C to U transitions in the SARS-CoV-2 mutants ranges from 40.7 to 65.2% (see Tables 4A–12A and Table 13) and suggests links to an mRNA-editing mechanism (Di Giorgio *et al*, 2020; Simmonds, 2020; Weber *et al*, 2020). It is unknown, how and when in the infection cycle cellular cytosine deaminases will interact with SARS-CoV-2 RNA to drive this mutagenic mechanism. Cellular apolipoprotein B mRNA-editing enzyme, catalytic polypeptide-like (APOBEC) would be a likely candidate to supply these enzymatic activities. The APOBEC class of mRNA-editing cytidine deaminases causes deamination of cytosine to uracil (Anant & Davidson, 2001). Moreover, the high incidence of C to U (T) transitions renders research on the occurrence of methyl-cytosine bases in SARS-CoV-2 RNA a project of considerable interest. Furthermore, the introduction of 14 point mutations and 3 small deletions in the genome of the B.1.1.7 also argues for mRNA editing as a plausible model. Interestingly, the APOBEC editing function has been interpreted as a cellular defense against intruding viral genomes. Hence, SARS-CoV-2 seems to exploit exactly this mechanism to promote its mutagenic potential.

This highly efficient cellular deamination mechanism raises the question of how the viral genome can be salvaged with time from a severe depletion of C/G bases. A screen for the occurrence of A or T to C or G exchanges among the mutations described here (Tables 3–12) reveals values of 9–23% that were identified in mutations from the UK, South Africa, the United States, India, and Germany. Obviously, there remain many unresolved questions about the mechanisms of viral mutagenesis.

### Will the constant selection of new mutants impinge upon the success of therapeutic or vaccination strategies?

There are multiple sources of vaccines against COVID-19 available now or at various stages of development, including those from Pfizer/BioNTec; AstraZeneca/Oxford University; Moderna/US National Institutes of Health; Johnson and Johnson Novavax; Curevac/Bayer and firms in Russia (Sputnik V), China, India, and many more. It is impossible to assess the vaccines' overall long-term efficacy against SARS-CoV-2 infections at this time. Vaccines available now have demonstrated a high level of clinical efficacy. There are also, however, preliminary data suggesting that evolution of viral variants may have diminished the efficacy of several vaccines against one of the new SARS-CoV-2 variants (https://www.janssen. com/johnson-johnson-announces-single-shot-janssen-covid-19-vacci ne-candidate-met-primary-endpoints; https://ir.novavax.com/news-releases/news-release-details/novavax-covid-19-vaccine-demonstra tes-893-efficacy-uk-phase-3; preprint: Wang *et al*, 2021; Xie *et al*, 2021). The emergence of novel variants and mutants of SARS-CoV-2 in short temporal succession (see Tables 1 and 2) and their difficult-to-assess impact on pathogenicity *in vivo* further complicate

predictions about future vaccine efficacy at this time. For example, some of the early laboratory assessments of efficacy versus the new variants have focused on neutralization by sera from immunized individuals; the clinical efficacy of vaccines, however, is likely to benefit from cell-mediated immunity as well (Burioni & Topol, 2021; Rubin & Baden, 2021). Although *in vitro* assessments of vaccine efficacy are important, the ultimate assessment of potency of a vaccine is clinical response (Burioni & Topol, 2021; Rubin & Baden, 2021). A recent medRxiv pre-print from Clalit Health Services, Israel's largest healthcare provider, offers very preliminary evidence that the South African variant has a higher rate of vaccine breakthrough than would be expected by its prevalence. Although not peer-reviewed and the numbers are small and the power limited, it stresses the need for vigilance and continued sequencing efforts. Moreover, sophisticated and specific plans are already in place to alter the COVID-19 vaccines to compensate for possible escape mutants. SARS-CoV-2 is a new and evolving pathogen. Effective vaccines have been developed within one year of the identification of the pathogen, a remarkably short time. Ingenuity and basic research are likely to offer solutions to help control the spread of SARS-CoV-2 and future emerging viruses.

### Limitations of this study

With mechanisms that complex and the speed at which new SARS-CoV-2 mutations keep arising, limitations on their study are inherent in this approach. For an ordered presentation of data, there had to be a cutoff in time. We chose January 20 and for data in Tables 1 and 2 March 2021 for the inclusion of new mutations. In addition, editorial work on the manuscript had to be considered. Of course, it was the principle of mutant development we were interested in and therefore had to compromise on the date and number of inclusions. We chose to address mutations isolated and sequenced from 10 different countries and were aware that in this way we missed interesting mutations elsewhere that had probably been selected under special environmental conditions. Moreover, there will undoubtedly be an unintended selection bias in that GISAID, our major source of documented sequences, may have concentrated on viral isolates causing the most severe forms of COVID-19. In fact, it will be a most rewarding topic for future research to correlate specific symptoms and/or severity of COVID-19 with the causative types of SARS-CoV-2 mutations. Most probably, it will become a demanding challenge to analyze mutants that will arise in response to the extensive worldwide programs of vaccinations against SARS-CoV-2.

## Materials and Methods

We analyzed complete SARS-CoV-2 genome sequences with known dates of sampling that were downloaded from GISAID, (i) complete sequences only were included. (ii) For a chosen time period, all complete sequences with a sampling date from each country were included. Sequences were binned according to sampling date. iii) Sequences by country were filtered by country using the GISAID interface (Shu & McCauley, 2017). Nucleotide sequences from the UK, South Africa, Brazil, the United States, India, Russia, France, Spain, Germany, and China were compared to the reference genome of the SARS-CoV-2 isolate from Wuhan-Hu-1, NCBI Reference

**The paper explained**

**Problem**

Upon extensive worldwide replication, SARS-CoV-2 mutants with increasing pathogenetic potential were rapidly selected. Details of viral mutagenesis and selection regimes are not understood. Vigorous vaccination programs against SARS-CoV-2 might be met in time by even more dangerous SARS-CoV-2 mutations.

**Results**

In several time intervals between January 2020 and March 2021, we inspected >383,500 complete SARS-CoV-2 RNA sequences from 10 different countries for the occurrence of mutations. In >1,700 sequences, the amino acid exchanges were also assigned. While up to April 2020, about 10 mutations were prevalent, the 77 to 100 new mutations expanded gradually in time intervals up to January 2021 when the complex variants of concern evolved in England, South Africa, and Brazil. Mutations were not confined to the spike protein but spanned the viral genome, and replacements rose up to 90% of RNA molecules. The disproportionate incidence of cytidine to uracil transitions might be due to cellular cytidine deaminases, possibly of the APOBEC type.

**Impact**

Our data document speed and efficiency of SARS-CoV-2 mutant selection that might gradually cause problems for therapeutic and vaccination programs. Viral mutant watch must go beyond the spike glycoprotein and include replication functions, the nucleocapsid phosphoprotein, and the poorly charted open reading frames of the viral genome.

Sequence: NC_045512.2. The programs Vector NTI Advance™ 11 (Invitrogen™), Tool Align X, or SnapGene (GSL Biotech), by using the algorithm MUSCLE (Multiple Sequence Comparison by Log-Expectation), for the alignment of sequences. Amino acid sequences were also analyzed with the program SnapGene. DNA sequence analyses of reverse transcripts of an RNA genome will have to be considered with the possibility that errors may have been introduced at several steps, e.g., by preferred reading mistakes of the reverse transcriptase due to specific sequence or structural properties of SARS-CoV-2 RNA. We have tried to overcome this obvious complication by analyzing a large number of genomes. Percentages were calculated by dividing the number of sequences with the mutation that were sampled at that time and available in the database by the total number of complete sequences with a known sampling date. In addition to the determination of mutants for defined time spans in ten countries, the total number of individual mutations was also determined in all sequences deposited to GISAID up until January 20, 2021, by using GESS (Global Evaluation of SARS-COV-2/hCOV-19 Sequences (Collier *et al*, 2021) as well as CoV-Glue (preprint: Singer *et al*, 2020) and PANGOLIN (Phylogenetic Assignment of Named Global Outbreak LINeages) https://github.com/hCoV-2019/pangolin) (Rambaut *et al*, 2020).

In the present study, somewhat arbitrarily, we set a 2% mark of mutations at a given nucleotide in the viral sequence as the cutoff for hotspot status and mutations recording in Tables 3–12. The SARS-CoV-2 RNA sequences investigated for mutant status had been deposited at time intervals of 2020 as follows:

Brazil: 02/25 to 08/15/2020; China-I: 12/23/2019 to 03/18/2020; China-II: 03/20 to 07/22/2020; France: April to 09/12/2020; Germany-I: February to 03/23/2020; Germany-II: February to 06/ 17/2020; Germany-III: 06/24 to 08/28/2020; Germany-IV 09/10 to 10/13; India: 01/27 to 05/27/2020 and 06/03 to 07/04/2020; Russia: 03/24 to 06/07/2020; South Africa: 09/01 to 12/07/2020; Spain: 06/ 01 to 09/20/2020; UK: 01/29–12/04/2020; US-I: 02/29 to 04/26/ 2020; US-II: 06/12 to 07/07/2020; US-III: 07/09 to 07/22/2020; and US-IV 08/01 to 12/01. Some of the data had been reported previously in Table 1 of Weber *et al*, 2020 (Weber *et al*, 2020), but were included here again for comparison. These data were designated with an asterisk.

## Data availability

The datasets produced in this study are available in the following databases: SARS-CoV-2 genome sequences: Global Initiative on Sharing Avian Influenza Data (https://www.gisaid.org/), SARS-CoV-2 genome sequences alignments: Google Drive (https://drive.google.com/drive/folders/1gWq1_jf2Seatl36KtalH7__8GHudOg5u?usp = sharing).

**Expanded View** for this article is available online.

## Acknowledgements

We acknowledge and are very grateful to the GISAID Initiative and for the hard work and open-science of the individual research laboratories and public health agencies that have made their genome data accessible on GISAID, on which this research is based. This research was supported by the Dr. Robert Pfleger Stiftung in Bamberg, Germany [5.12.2018]. W.D. is indebted to the Institute for Clinical and Molecular Virology of FAU in Erlangen, Germany, for their continued hospitality extended to the Epigenetics Group. Open Access funding enabled and organized by Projekt DEAL.

## Author contributions

SW carried out all work involving sequence selection and formal analyses and was involved in the conceptualization of the project and in the analysis and interpretation of data. CMR performed the analysis on the large sequence database and variants of interest/concern using GISAID, GESS, CoV-Glue, and other computational tools, statistical analyses, interpretation of the data, and writing of the manuscript. BW and HB contributed to the analysis and interpretation of the data. BW contributed to writing the manuscript. WD initiated the project, was involved in the conceptualization of the project, and in the analysis and interpretation of data and wrote the manuscript with CMR's and BW's contributions.

## Conflict of interest

The authors declare that they have no conflict of interest.

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
