## [Review Process File · EMBO Molecular Medicine]

Rev.EMM-2021-14062 SARS-CoV-2 Worldwide Replication Drives Rapid Rise and Selection of Mutations:

Stefanie Weber, Christina Ramirez, Barbara Weiser, Harold Burger, and Walter Doerfler
DOI: [10.15252/emmm.202114062](https://doi.org/10.15252/emmm.202114062)

Corresponding author: Walter Doerfler (walter.doerfler@t-online.de)

Review Timeline:

Submission Date:	2nd Feb 21
Editorial Decision:	24th Mar 21
Revision Received:	13th Apr 21
Editorial Decision:	14th Apr 21
Revision Received:	18th Apr 21
Accepted:	20th Apr 21

Editor: Zeljko Durdevic

Transaction Report:

24th Mar 2021

Dear Prof. Doerfler,

Thank you for the submission of your manuscript to EMBO Molecular Medicine, and please accept my apologies for the delay in getting back to you. We have received feedback from two of the three reviewers who agreed to evaluate your manuscript. Should referee #3 provide a report, we will send it to you, with the understanding that we will not ask for an additional revision. As you will see from the reports below, both referees find the study interesting and important. However, referee #2 also raises important criticism that I would like you to address in a major revision of the current manuscript.

EMBO Molecular Medicine encourages a single round of revision only and therefore, acceptance or rejection of the manuscript will depend on the completeness of your responses included in the next, final version of the manuscript. For this reason, and to save you from any frustrations in the end, I would strongly advise against returning an incomplete revision.

We would welcome the submission of a revised version within three months for further consideration. However, we realize that the current situation is exceptional on the account of the COVID-19/SARS-CoV-2 pandemic. Please let us know if you require longer to complete the revision.

I look forward to receiving your revised manuscript.

Yours sincerely,

Zeljko Durdevic

***** Reviewer's comments *****

Referee #1 (Remarks for Author):

The important manuscript by Weber, Ramirez, Weiser, Burger and Doerfler builds substantially on their initial publication in September 2020 focused on tracking SARS-CoV2 variants arising in 10 different countries representing 5 continents as their genomes become fully sequenced, right up to date through January 20, 2021 and deposited in the Global Initiative of Sharing All Influenza Data - GISAID. In this report, they analyzed over 383,000 complete viral genomes deposited in the global database from 10 countries that are carrying out considerable sequencing of isolates from patients.

Starting from a baseline of the original SARS-CoV2 isolate from Wuhan, China, a major objective of this investigations was to track the emergence of amino acid sequence variants throughout the viral genome and look for identities, similarities and differences in various populations around the world with goals of distinguishing independent emergence and positive selection of certain mutations as well as transmission of genetically identifiable lineages by travelers between the

nations, near and far. Some of the original "wild type" sequences were observed to "dropout" in selected viral populations over time, indicative of insufficient fitness relative to rapidly emerging variants. Some of these are especially worrisome from a public health and containment perspective because they can negatively impact PCR and other screening tests, while other "escape mutations" can diminish the neutralizing ability of the vaccines that are now becoming available, notably the Oxford/AstraZeneca vaccine against the rapidly emerging South African variant.

This investigation is most timely, expansive and inclusive, as well as exceptionally granular, as presented in a large number of Tables and associated data files. By limiting the data set to viral genomes fully sequenced, and by emphasizing reproducible sites of variation to deemphasize the inevitable one-off random sequencing errors, the authors carefully curated the data sets to focus on evolutionary patterns of changing viral fitness over the full year of the global pandemic. The major sites of change observed were most often in the Spike gene but also significantly in a number of genes associated with viral replication and packaging, all of which are conceptually consistent with successful attachment of infection virions to target cells, rapid and robust amplification, efficient packaging, and short infection cycle turn-around. Fairly often, the variant replacement often rose to 50% upwards to over 90%, relative to the dominant sequence just months previously, so the selection can be quite rapid. Interestingly, some significant differences in mutant sites were seen continent to continent, implying ethnic and socio-economic factors were at play in the selection for SARS-CoV2 fitness. The authors took particular note of the types of mutation, especially those that resulted in non-synonymous changes in the chemical nature of the replacement amino acid. These results will inspire and guide protein biophysical chemists and enzymologists to investigate the molecular changes and their mechanistic consequences. A most notable consistency was the very disproportionate preference for cytidine to uracil (thymidine) transitions across viral genes and human populations. This points to deamination of cytosine, which has been linked to the APOBEC protein, an RNA editing factor. Intriguingly, the SARS-CoV2 cell surface receptor ACE interacts with APOBEC. The details of these suggestively linked phenomena await future mechanistic research. Nonetheless, it may be that this corona virus co-ops a cellular antiviral defense to establish an environment where the virus can potentially benefit from occasional mutagenesis, noting that SARS-CoV2 RNA replication has its own proof-reading capabilities so is not as likely as most RNA viruses to generate variants in the course of genomic replication.

The manuscript is especially clearly written. I note only a single confusing typographical problem in line 2 of the top of page 17, apparently due to the insertion of a series of country names. I am not convinced of the relevance or value of the two parenthetical notes added during submission.

The comprehensive and rather easy to navigate data in this report will be a most valuable resource for virologists, epidemiologists and global public health agencies and practitioners as well as commercial providers of screening tests or vaccines who continue to monitor and react to the emergence of variants. This manuscript should be accepted for rapid publication with enthusiasm. Notably, I strongly recommend that the authors be accorded the rather unusual opportunity to update their database now and over the coming months should they choose to do so. They are to be congratulated on a Herculean task of consolidating such a vast database into a functional format.

Referee #2 (Comments on Novelty/Model System for Author):

This is an exhaustive characterization of the emergence of SARS-CoV-2 mutants worldwide, using standard bioinformatic pipelines and an open database of SARS-CoV-2 sequences. It is technically well done, and may have high medical impact. The use of pipelines and GISAID is not novel, though the spatiotemporal evaluation of mutations in this exhaustive manner is novel.

Referee #2 (Remarks for Author):

This is an exhaustive evaluation of the emergence of SARS-COV-2 mutants over time, from about 10 predominant mutants to May 2020, followed by an increase in the number of new mutants from June 2020 to December 2020. The work in general is a straightforward characterization of the emergence of these mutations in various countries, and should be useful as a reference for these mutations.

In a manuscript this exhaustive, certain statements are made that require further support or modification.

(1) The authors postulate APOBEC mRNA editing enzymes should be active in the emergence of these mutations, given the high number of C to U transitions. The authors take this a step further, and postulate that ACE2 binding of the spike has something to do with this process. While an interesting speculation, there is nothing provided in the manuscript by literature or possible mechanism linking APOBEC to ACE2. This needs further explanation or modification.

(2) The authors do state briefly that synonymous mutation may have effects outside of protein modification, such as changes in mRNA secondary structure. Given the number of synonymous mutations found, this explanation should be expanded, and possibly include codon choice and translation efficiency as one possible explanation for differences in viral transmissibility and pathogenesis.

(3) A separate section on limitations of this study should be included, especially for the non-specialist in this field. In particular, selection bias in sequencing should be emphasized. This is important in that viral sequences may have been provided to GISAID from patients with the most severe disease, which may give an inaccurate picture of viral mutations in less severely affected patients.

(4) The choice of 2% cut off for mutation prevalence to report is as the authors claim arbitrary. This choice should be further justified.

(5) There is a large literature on viral evolution when viruses enter novel hosts, starting with the analysis of the MYXV virus in rabbits in Australia in the 1950s. Examination of this literature, with reference to it in the manuscript, would potentially be helpful in explaining the emergence of mutations in SARS CoV 2 in humans.

(6) Page 16 at the top: "High current incidence of Covid-19 is paralleled by high numbers of new mutations and variants, although this relationship was not observed in Brazil or Russia." Can the authors provide a possible explanation?

(7) Page 17: The link between ACE2 binding and APOBEC enzyme activation is unsupported. Additionally, the authors fail to note that SARS-CoV-2 may have multiple accessory factors for entry, such as CD147, NRP-1, and DC-SIGN among others. How do the mutations affect binding to

these accessory factors, and do they have a link to APOBEC enzymes? This section is highly speculative, and need better literature support for its assertions.

(8) Page 17: This paragraph at the bottom, discussing the impact of variants on vaccine efficacy, while interesting, is cursory for what is turning out to be a very complicated topic. Perhaps it may be best to reduce or eliminate the part of the paragraph beginning with "The emergence..." since this is speculation.

(9) The note added during submission at the top of page 18 references a very speculative report, and I am not sure it belongs in this manuscript, which should focus on the categorization of mutations through time.

(10) I would eliminate the last paragraph of the manuscript "Conceptual bridge.." in that while interesting, it does not add much for the general reader, and the analogies are a bit confusing.

Reviewers' Reports and our Point-by-Point Response (Response in yellow)**Referee #1 (Remarks for Author)**

The important manuscript by Weber, Ramirez, Weiser, Burger and Doerfler builds substantially on their initial publication in September 2020 focused on tracking SARS-CoV2 variants arising in 10 different countries representing 5 continents as their genomes become fully sequenced, right up to date through January 20, 2021 and deposited in the Global Initiative of Sharing All Influenza Data - GISAID. In this report, they analyzed over 383,000 complete viral genomes deposited in the global database from 10 countries that are carrying out considerable sequencing of isolates from patients.

Starting from a baseline of the original SARS-CoV2 isolate from Wuhan, China, a major objective of this investigations was to track the emergence of amino acid sequence variants throughout the viral genome and look for identities, similarities and differences in various populations around the world with goals of distinguishing independent emergence and positive selection of certain mutations as well as transmission of genetically identifiable lineages by travelers between the nations, near and far. Some of the original "wild type" sequences were observed to "dropout" in selected viral populations over time, indicative of insufficient fitness relative to rapidly emerging variants. Some of these are especially worrisome from a public health and containment perspective because they can negatively impact PCR and other screening tests, while other "escape mutations" can diminish the neutralizing ability of the vaccines that are now becoming available, notably the Oxford/AstraZeneca vaccine against the rapidly emerging South African variant.

This investigation is most timely, expansive and inclusive, as well as exceptionally granular, as presented in a large number of Tables and associated data files. By limiting the data set to viral genomes fully sequenced, and by emphasizing reproducible sites of variation to deemphasize the inevitable one-off random sequencing errors, the authors carefully curated the data sets to focus on evolutionary patterns of changing viral fitness over the full year of the global pandemic. The major sites of change observed were most often in the Spike gene but also significantly in a number of genes associated with viral replication and packaging, all of which are conceptually consistent with successful attachment of infection virions to target cells, rapid and robust amplification, efficient packaging, and short infection cycle turn-around. Fairly often, the variant replacement often rose to 50% upwards to over 90%, relative to the dominant sequence just months previously, so the selection can be quite rapid. Interestingly, some significant differences in mutant sites were seen continent to continent, implying ethnic and socio-economic factors were at play in the selection for SARS-CoV2 fitness. The authors took particular note of the types of mutation, especially those that resulted in non-synonymous changes in the chemical nature of the replacement amino acid. These results will inspire and guide protein biophysical chemists and enzymologists to investigate the molecular changes and their mechanistic consequences. A most notable consistency was the very disproportionate preference for cytidine to uracil (thymidine) transitions across viral genes and human populations. This points to deamination of cytosine, which has been linked to the APOBEC protein, an RNA editing factor. Intriguingly, the SARS-CoV2 cell surface receptor ACE interacts with APOBEC. The details of these suggestively linked phenomena await future mechanistic research. Nonetheless, it may be that this corona virus co-ops a cellular antiviral defense to establish an environment where the virus can potentially benefit from occasional mutagenesis, noting that SARS-CoV2 RNA replication has its own proof-reading capabilities so is not as likely as most RNA viruses to generate variants in the course of genomic replication.

The manuscript is especially clearly written. I note only a single confusing typographical problem in line 2 of the

top of page 17, apparently due to the insertion of a series of country names. I am not convinced of the relevance or value of the two parenthetical notes added during submission.

Response:

1. As will also be explained in response to items (1) and (7) raised by referee #2, the remark on a hypothetical association between the ACE-2 receptor and the deaminase functions of APOBEC has been deleted. Text passages on pages 3 (Synopsis) and 17 (second paragraph) have been completely rewritten. Now, we just mention that cellular deaminases, possibly of the APOBEC type, might be responsible for the frequent C to T transitions
2. On page 18, the “Note added during submission” has been removed. Similarly, the typographical problem now on page 17, first paragraph, lines 3 to 4 from the bottom – ~~UK, South Africa, Brazil, ...~~ – that had been added inadvertently - has been amended. We have updated the data bases in Tables 1 and 2 as of March 31, 2021. In the interest of not delaying publication, we decided to forego the referee’s offer to update additional data bases. We thank you for the opportunity. Hence Tables 1 and 2 have been revised. In contrast Tables 3 to 12B have not been altered and will be resubmitted in the original (February 1) format.
3. The updated numbers of variant incidence have also been added to the description of mutants arising in individual countries. Please see pages 7 (UK), 8 (South Africa), 9 (Brazil), 10 (US), 11 (India), 12 (Russia and France), 14 (Spain and Germany), 15 (China, only a comment).

The comprehensive and rather easy to navigate data in this report will be a most valuable resource for virologists, epidemiologists and global public health agencies and practitioners as well as commercial providers of screening tests or vaccines who continue to monitor and react to the emergence of variants. This manuscript should be accepted for rapid publication with enthusiasm. Notably, I strongly recommend that the authors be accorded the rather unusual opportunity to update their database now and over the coming months should they choose to do so. They are to be congratulated on a Herculean task of consolidating such a vast database into a functional format.

Referee #2 (Comments on Novelty/Model System for Author):

This is an exhaustive characterization of the emergence of SARS-CoV-2 mutants worldwide, using standard bioinformatic pipelines and an open database of SARS-CoV-2 sequences. It is technically well done, and may have high medical impact. The use of pipelines and GISAID is not novel, though the spatiotemporal evaluation of mutations in this exhaustive manner is novel.

Referee #2 (Remarks for Author):

This is an exhaustive evaluation of the emergence of SARS-COV-2 mutants over time, from about 10 predominant mutants to May 2020, followed by an increase in the number of new mutants from June 2020 to December 2020. The work in general is a straightforward characterization of the emergence of these mutations in various countries, and should be useful as a reference for these mutations.

In a manuscript this exhaustive, certain statements are made that require further support or modification.

- (1) The authors postulate APOBEC mRNA editing enzymes should be active in the emergence of these

mutations, given the high number of C to U transitions. The authors take this a step further, and postulate that ACE2 binding of the spike has something to do with this process. While an interesting speculation, there is nothing provided in the manuscript by literature or possible mechanism linking APOBEC to ACE2. This needs further explanation or modification.

Response:

Items (1) and (7). As mentioned above (item 1 of referee #1), the remark on a hypothetical association between the ACE-2 receptor and the deaminase functions of APOBEC has been deleted. Text passages on pages 3 (Synopsis) and 17 (second paragraph) have been completely rewritten. Now, we just mention that cellular deaminases, possibly of the APOBEC type, might be responsible for the frequent C to T transitions.

(2) The authors do state briefly that synonymous mutation may have effects outside of protein modification, such as changes in mRNA secondary structure. Given the number of synonymous mutations found, this explanation should be expanded, and possibly include codon choice and translation efficiency as one possible explanation for differences in viral transmissibility and pathogenesis.

Response:

As recommended by the reviewer, we have expanded this discussion on page 13/14.

- (3) A separate section on limitations of this study should be included, especially for the non-specialist in this field. In particular, selection bias in sequencing should be emphasized. This is important in that viral sequences may have been provided to GISAID from patients with the most severe disease, which may give an inaccurate picture of viral mutations in less severely affected patients.

Response:

Following up on this recommendation, a new section has been added on page 19: (v) “*Limitations of this study*” in which a number of items, including the ones suggested by referee #2, have been discussed. Factors that might have led to selection bias in the choice of analyzed sequences have also been discussed on page 5, last paragraph.

- (4) The choice of 2% cut off for mutation prevalence to report is as the authors claim arbitrary. This choice should be further justified.

Response:

We have added an explanation on page 6, third paragraph:

Of course, it can be argued that a cut off for the registration of mutants at 2% incidence is arbitrary. However, we cannot predict with certainty which mutations at low incidence of occurrence at present will become more predominant in the future during rapid worldwide viral replication in the current pandemic. A feasible strategy will be to install mutant watch programs and remain on the alert for the rise of new mutations. This strategy can be implemented only by highly efficient SARS-CoV-2 RNA sequencing strategies that will have to be instituted as widely as possible and

without delay. Moreover, 2% was also chosen for publication practicality reasons as well; due to the table lengths becoming very large when using 1% as a cutoff. We thought that 2% was the right balance in being able to flag rare mutations that may become predominant in the future in several countries with readability for interested researchers.

(5) There is a large literature on viral evolution when viruses enter novel hosts, starting with the analysis of the MYXV virus in rabbits in Australia in the 1950s. Examination of this literature, with reference to it in the manuscript, would potentially be helpful in explaining the emergence of mutations in SARS CoV 2 in humans.

Response:

Viral evolution is definitely a most interesting topic. In the context of this study, we can consider it, however, only to a limited extent. Accordingly, we have added paragraph 2 on page 16 plus two new references (MacLean et al, 2021) and [WHO-2019-nCoV-FAQ-Virus_origin-2020.1-eng.pdf (122.6K)].

(6) Page 16 at the top: "High current incidence of Covid-19 is paralleled by high numbers of new mutations and variants, although this relationship was not observed in Brazil or Russia." Can the authors provide a possible explanation?

Response:

We surmise that the relevant data on this point have not been available. A short remark has been added on page 16, first paragraph.

(7) Page 17: The link between ACE2 binding and APOBEC enzyme activation is unsupported. Additionally, the authors fail to note that SARS-CoV-2 may have multiple accessory factors for entry, such as CD147, NRP-1, and DC-SIGN among others. How do the mutations affect binding to these accessory factors, and do they have a link to APOBEC enzymes? This section is highly speculative, and need better literature support for its assertions.

Response: Please, see response to item (1)

(8) Page 17: This paragraph at the bottom, discussing the impact of variants on vaccine efficacy, while interesting, is cursory for what is turning out to be a very complicated topic. Perhaps it may be best to reduce or eliminate the part of the paragraph beginning with "The emergence..." since this is speculation.

Response:

We want to keep and consider this discussion, now on page 18 under „iv [*Will the constant selection of new mutants impinge upon the success of therapeutic or vaccination strategies?*]. In the context of present reports on the spread of constantly evolving new variants like B.1.1.7, B.1.135, P1, B.1.429 + B.1.427, B.1.525E and possibly others (see also the updated versions of Tables 1 and 2) this part of the manuscript is of high actual interest and describes the current state of the Covid-19 pandemic realistically. New variants of concern and mutations with yet un-identified pathogenicity might surface as SARS-CoV-2 continues to replicate actively in many parts of the world. We therefore, with due respect, should like to keep this section. Thank you.

(9) The note added during submission at the top of page 18 references a very speculative report, and I am not sure it belongs in this manuscript, which should focus on the categorization of mutations through time.

Response:

The note added during submission now on page 18 together with the reference were removed from the manuscript. ~~Domingo and Perales [J. Virol. 2021, doi: 10.1128/JVI.02437-20~~

(10) I would eliminate the last paragraph of the manuscript "Conceptual bridge.." in that while interesting, it does not add much for the general reader, and the analogies are a bit confusing.

Response:

~~B-H, Zhong E, Berger B, Bryson B. Learning the language of viral evolution and escape. Science 2021;374~~ Indeed, an interesting paper. However, we agree that the section "Conceptual Bridge" and the accompanying reference now on page 19 would not add much and has been removed from the manuscript.

In addition:

In accordance with journal style, the Materials and Methods section has now been moved to pages 19 to 20 (top), including the comment on Data Availability. There are no changes in this section.

Minor numerical corrections had to be made in Table 13 (highlighted yellow). All calculations have now been based on the results presented in the A Tables. Minor discrepancies, now corrected, arose because we had based some of the calculations on the data in the B Tables.

A "four bullet points" synopsis (page 3, also submitted separately) of the main findings of our study have been incorporated into the manuscript as well as the "Paper explained" statements (page 21).

Comments on items 1) to 15) from pages 2/3 in the Decision Letter

- 1) Revised manuscript has been formatted .docx. All changes have been highlighted yellow.
- 2) Quality in TIFF has been included in the submission.
- 3) This letter includes both the reviewers' reports and our point-by-point responses (in yellow) that have been intercalated with the referees' comments.
- 4) A completed author checklist will be provided as a separate file.
- 5) Walter Doerfler's ORCID ID is: orcid.org/0000-0002-9971-0138
- 6) Datasets have been deposited as described on page 20 under **Data availability**
- 7) Not required.
- 8) Data citations in reference list have been converted to journal style.
- 9) There is no Supplementary Information for this manuscript.
- 10) "Paper explained" section can be found on page 21 of the manuscript.
- 11) Not relevant.

- 12) Author contributions have been described on page 20 of the manuscript.
- 13) Conflict of Interests statement is now on page 21 of the manuscript.
- 14) A Synopsis has been added on page 3 and has also been submitted separately.
Moreover, Figure 1 has been added as a TIFF file which could be used as **visual abstract**.
- 15) We understand the journal to provide that part.

Finally, we wish to thank the referees and you for your constructive comments. We understand that the revised manuscript will now be acceptable for publication.

Best wishes and regards,

Walter Doerfler – for all authors

Professor *emeritus*, Molecular Genetics, Cologne, Guest Professor, Virology, FAU Erlangen

14th Apr 2021

Dear Prof. Doerfler,

Thank you for the submission of your revised manuscript to EMBO Molecular Medicine. I am pleased to inform you that we will be able to accept your manuscript pending the following final amendments:

- 1) Tables: Tables should be uploaded as one file per table. Where there are several parts that don't fit onto one page, they could be combined as tabs in an excel table. Main tables are typeset and can only be displayed in black and white. Please remove the red colour from some of the tables and replace it with bolded font.
- 2) In the main manuscript file, please do the following:
 - Correct/answer the track changes suggested by our data editors by working from the attached/uploaded document.
 - Please shorten keywords to max. 5.
 - Remove text highlight colour.
 - Add callouts for Figure 1 and Table 2.
- 3) Synopsis: Every published paper now includes a 'Synopsis' to further enhance discoverability. Synopses are displayed on the journal webpage and are freely accessible to all readers. They include separate synopsis image and synopsis text.
 - Synopsis image: Please provide a striking image or visual abstract as a high-resolution jpeg file 550 px-wide x (250-400)-px high to illustrate your article.
- 4) For more information: There is space at the end of each article to list relevant web links for further consultation by our readers. Could you identify some relevant ones and provide such information as well? Some examples are patient associations, relevant databases, OMIM/proteins/genes links, author's websites, etc...
- 5) Press release: Please inform us as soon as possible and latest at the time of submission of the revised manuscript if you plan a press release for your article so that our publisher could coordinate publication accordingly.
- 6) Please be aware that we use a unique publishing workflow for COVID-19 papers: a non-typeset PDF of the accepted manuscript is published as "Just Accepted" on our website. With respect to a possible press release, we have the option to not post the "Just Accepted" version if you prefer to wait with the press release for the typeset version. Please let us know whether you agree to publication of a "Just accepted" version or you prefer to wait for the typeset version.
- 7) As part of the EMBO Publications transparent editorial process initiative (see our Editorial at <http://embomolmed.embopress.org/content/2/9/329>), EMBO Molecular Medicine will publish online a Review Process File (RPF) to accompany accepted manuscripts. This file will be published in conjunction with your paper and will include the anonymous referee reports, your point-by-point response and all pertinent correspondence relating to the manuscript. Let us know whether you agree with the publication of the RPF and as here, if you want to remove or not any figures from it prior to publication. Please note that the Authors checklist will be published at the end of the RPF.
- 8) Please provide a point-by-point letter INCLUDING my comments as well as the reviewer's reports and your detailed responses (as Word file).

I look forward to reading a new revised version of your manuscript as soon as possible.

Yours sincerely,

Zeljko Durdevic

The authors performed the requested editorial changes.

We are pleased to inform you that your manuscript is accepted for publication and is now being sent to our publisher to be included in the next available issue of EMBO Molecular Medicine.

Walter Doerfler

EMBO Molecular Medicine

EMM-2021-14062